# Learning Zero-Shot Cooperation with Humans, Assuming Humans Are Biased

**Chao Yu**[1*], **Jiaxuan Gao**[1,3*], **Weilin Liu**[1], **Botian Xu**[3], **Hao Tang**[1], **Jiaqi Yang**[2], **Yu Wang**[1†], **Yi Wu**[1,3†]

[1] Tsinghua University, [2] UC Berkeley, [3] Shanghai Qi Zhi Institute

zoeyuchao@gmail.com

## Abstract

There is a recent trend of applying multi-agent reinforcement learning (MARL) to train an agent that can cooperate with humans in a zero-shot fashion without using any human data. The typical workflow is to first repeatedly run self-play (SP) to build a policy pool and then train the final adaptive policy against this pool. A crucial limitation of this framework is that every policy in the pool is optimized w.r.t. the environment reward function, which implicitly assumes that the testing partners of the adaptive policy will be precisely optimizing the same reward function as well. However, human objectives are often substantially biased according to their own preferences, which can differ greatly from the environment reward. We propose a more general framework, *Hidden-Utility Self-Play* (HSP), which explicitly models human biases as hidden reward functions in the self-play objective. By approximating the reward space as linear functions, HSP adopts an effective technique to generate an augmented policy pool with biased policies. We evaluate HSP on the *Overcooked* benchmark. Empirical results show that our HSP method produces higher rewards than baselines when cooperating with learned human models, manually scripted policies, and real humans. The HSP policy is also rated as the most assistive policy based on human feedback.

## 1 Introduction

Building intelligent agents that can interact with, cooperate and assist humans remains a long-standing AI challenge with decades of research efforts (Klien et al., 2004; Ajoudani et al., 2018; Dafoe et al., 2021). Classical approaches are typically model-based, which (repeatedly) build an effective behavior model over human data and plan with the human model (Sheridan, 2016; Carroll et al., 2019; Bobu et al., 2020). Despite great successes, this model-based paradigm requires an expensive and time-consuming data collection process, which can be particularly problematic for complex problems tackled by today's AI techniques (Kidd & Breazeal, 2008; Biondi et al., 2019) and may also suffer from privacy issues (Pan et al., 2019).

Recently, multi-agent reinforcement learning (MARL) has become a promising approach for many challenging decision-making problems. Particularly in competitive settings, AIs developed by MARL algorithms based on self-play (SP) defeated human professionals in a variety of domains (Silver et al., 2018; Vinyals et al., 2019; Berner et al., 2019). This empirical evidence suggests a new direction of developing strong AIs that can directly cooperate with humans in a similar "model-free" fashion, i.e., via self-play.

Different from zero-sum games, where simply adopting a Nash equilibrium strategy is sufficient, an obvious issue when training cooperative agents by self-play is *convention overfitting*. Due to the existence of a large number of possible optimal strategies in a cooperative game, SP-trained agents can easily converge to a particular optimum and make decisions solely based on a specific behavior pattern, i.e., *convention* (Lowe et al., 2019; Hu et al., 2020), of its co-trainers, leading to poor generalization ability to unseen partners. To tackle this problem, recent works proposed a two-staged framework by first developing a diverse policy pool consisting of multiple SP-trained policies, which possibly cover different conventions, and then further training an adaptive policy against this policy pool (Lupu et al., 2021; Strouse et al., 2021; Zhao et al., 2021).

Despite the empirical success of this two-staged framework, a fundamental drawback exists. Even though the policy pool prevents convention overfitting, each SP-trained policy in the pool remains a solution, which is either optimal or sub-optimal, to a fixed reward function specified by the underlying cooperative game. This implies a crucial generalization assumption that any test-time partner

---

*Equal Contribution
†Equal Advising

will be *precisely* optimizing the specified game reward. Such an assumption results in a pitfall in the case of cooperation with *humans*. Human behavior has been widely studied in cognitive science (Griffiths, 2015), economics (Wilkinson & Klaes, 2017) and game theory (Fang et al., 2021). Systematic research has shown that humans' utility functions can be substantially biased even when a clear objective is given (Pratt, 1978; Selten, 1990; Camerer, 2011; Barberis, 2013), suggesting that human behaviors may be subject to an unknown reward function that is very different from the game reward (Nguyen et al., 2013). This fact reveals an algorithmic limitation of the existing SP-based methods.

In this work, we propose *Hidden-Utility Self-Play* (HSP), which extends the SP-based two-staged framework to the assumption of biased humans. HSP explicitly models the human bias via an additional hidden reward function in the self-play training objective. Solutions to such a generalized formulation are capable of representing any non-adaptive human strategies. We further present a tractable approximation of the hidden reward function space and perform a random search over this approximated space when building the policy pool in the first stage. Hence, the enhanced pool can capture a wide range of possible human biases beyond conventions (Hu et al., 2020; Zhao et al., 2021) and skill-levels (Dafoe et al., 2021) w.r.t. the game reward. Accordingly, the final adaptive policy derived in the second phase can have a much stronger adaptation capability to unseen humans.

We evaluate HSP in a popular human-AI cooperation benchmark, *Overcooked* (Carroll et al., 2019), which is a fully observable two-player cooperative game. We conduct comprehensive ablation studies and comparisons with baselines that do not explicitly model human biases. Empirical results show that HSP achieves superior performances when cooperating with behavior models learned from human data. In addition, we also consider a collection of manually scripted biased strategies, which are ensured to be sufficiently distinct from the policy pool, and HSP produces an even larger performance improvement over the baselines. Finally, we conduct real human studies. Collected feedbacks show that the human participants consistently feel that the agent trained by HSP is much more assistive than the baselines. We emphasize that, in addition to algorithmic contributions, our empirical analysis, which considers learned models, script policies and real humans as diverse testing partners, also provides a more thorough evaluation standard for learning human-assistive AIs.

## 2 RELATED WORK

There is a broad literature on improving the zero-shot generalization ability of MARL agents to unseen partners (Kirk et al., 2021). Particularly for cooperative games, this problem is often called *ad hoc team play* (Stone et al., 2010) or *zero-shot cooperation* (ZSC) (Hu et al., 2020). Since most existing methods are based on self-play (Rashid et al., 2018; Yu et al., 2021), how to avoid convention overfitting becomes a critical challenge in ZSC. Representative works include improved policy representation (Zhang et al., 2020; Chen et al., 2020), randomization over invariant game structures (Hu et al., 2020; Treutlein et al., 2021), population-based training (Long* et al., 2020; Lowe* et al., 2020; Cui et al., 2021) and belief modeling for partial observable settings (Hu et al., 2021; Xie et al., 2021). *Fictitious co-play* (FCP) (Strouse et al., 2021) proposes a two-stage framework by first creating a pool of self-play policies and their previous versions and then training an adaptive policy against them. Some techniques improves the diversity of the policy pool (Garnelo et al., 2021; Liu et al., 2021; Zhao et al., 2021; Lupu et al., 2021) for a stronger adaptive policy (Knott et al., 2021).

We follow the FCP framework and augment the policy pool with biased strategies. Notably, techniques for learning a robust policy in competitive games, such as policy ensemble (Lowe et al., 2017), adversarial training (Li et al., 2019) and double oracle (Lanctot et al., 2017), are complementary to our focus.

Building AIs that can cooperate with humans remains a fundamental challenge in AI (Dafoe et al., 2021). A critical issue is that humans can be systematically biased (Camerer, 2011; Russell, 2019). Hence, great efforts have been made to model human biases, such as irrationality (Selten, 1990; Bobu et al., 2020; Laidlaw & Dragan, 2022), risk aversion (Pratt, 1978; Barberis, 2013), and myopia (Evans et al., 2016). Many popular models further assume humans have hidden subject utility functions (Nguyen et al., 2013; Hadfield-Menell et al., 2016; Eckersley, 2019; Shah et al., 2019). Conventional methods for human-AI collaboration require an accurate behavior model over human data (Ajoudani et al., 2018; Kwon et al., 2020; Kress-Gazit et al., 2021; Wang et al., 2022), while we consider the setting of no human data. Hence, we explicitly model human biases as a hidden utility function in the self-play objective to reflect possible human biases beyond conventions w.r.t. optimal rewards. We prove that such a hidden-utility model can represent any strategy of non-adaptive humans. Notably, it is also feasible to generalize our model to capture higher cognitive hierarchies (Camerer et al., 2004), which we leave as a future direction.

We approximate the reward space by a linear function space over event-based features. Such a linear representation is typical in inverse reinforcement learning (Ng & Russell, 2000), policy trans-

fer (Barreto et al., 2017b), evolution computing (Cully et al., 2015) and game theory (Winterfeldt & Fischer, 1975; Kiekintveld et al., 2013). Event-based rewards are also widely adopted as a general design principle in robot learning (Fu et al., 2018; Zhu et al., 2019; Ahn et al., 2022). We perform randomization over feature weights to produce diverse biased strategies. Similar ideas have been adopted in other settings, such as generating adversaries (Paruchuri et al., 2006), emergent team-formation (Baker, 2020), and searching for diverse Nash equilibria in general-sum games (Tang et al., 2020). In our implementation, we use multi-reward signals as an approximate metric to filter out duplicated policies, which is inspired by the quality diversity method (Pugh et al., 2016). There are also some works utilizing model-based methods to solve zero-shot cooperation Wu et al. (2021). Their focus is orthogonal to our approach since they focus more on constructing an adaptive agent, while our approach aims to find more diverse strategies. Besides, we adopt an end-to-end fashion to train an adaptive agent, which is more general. Lastly, our final adaptive agent assumes a zero-shot setting without any data from its testing partner. This can be further extended by allowing meta-adaptation at test time (Charakorn et al., 2021; Gupta et al., 2021; Nekoei et al., 2021), which we leave as a future direction.

## 3 PRELIMINARY

**Two-Player Human-AI Cooperative Game:** A human-AI cooperative game is defined on a world model, i.e., a two-player Markov decision process denoted by $M = \langle \mathcal{S}, \mathcal{A}, P, R \rangle$, with one player with policy $\pi_A$ being an AI and the other with policy $\pi_H$ being a human. $\mathcal{S}$ is a set of world states. $\mathcal{A}$ is a set of possible actions for each player. $P$ is a transition function over states given the actions from both players. $R$ is a global reward function. A policy $\pi_i$ produces an action $a_t^{(i)} \in \mathcal{A}$ given a world state $s_t \in \mathcal{S}$ at the time step $t$. We use the expected discounted return $J(\pi_A, \pi_H) = \mathbb{E}_{s_t, a_t^{(i)}} \left[ \sum_t \gamma^t R(s_t, a_t^{(A)}, a_t^{(H)}) \right]$ as the objective. Note that $J(\pi_H, \pi_A)$ can be similarly defined, and we use $J(\pi_A, \pi_H)$ for conciseness without loss of generality. Let $P_H : \Pi \to [0, 1]$ be the unknown distribution of human policies. The goal is to find a policy $\pi_A$ that maximizes the expected return with an unknown human, i.e., $\mathbb{E}_{\pi_H \sim P_H}[J(\pi_H, \pi_A)]$. In practice, many works construct or learn a policy distribution $\hat{P}_H$ to approximate real-world human behaviors, leading to an approximated objective for $\pi_A$, i.e., $\mathbb{E}_{\hat{\pi}_H \sim \hat{P}_H}[J(\pi_A, \hat{\pi}_H)]$.

**Self-Play for Human-AI Cooperation:** Self-play (SP) optimizes $J(\pi_1, \pi_2)$ with two parametric policies $\pi_1$ and $\pi_2$ and takes $\pi_1$ as $\pi_A$ without use of human data. However, SP suffers from poor generalization since SP converges to a specific optimum and overfits the resulting behavior convention. Population-based training (PBT) improves SP by representing $\pi_i$ as a mixture of $K$ individual policies $\{\pi_i^{(k)}\}_{k=1}^K$ and runs *cross-play* between policies by optimizing the expected return (Long* et al., 2020; Lowe* et al., 2020; Cui et al., 2021). PBT can be further improved by adding a diversity bonus over the population (Garnelo et al., 2021; Liu et al., 2021; Lupu et al., 2021).

**Fictitious Co-Play (FCP):** FCP (Strouse et al., 2021) is a recent work on zero-shot human-AI cooperation with strong empirical performances. FCP extends PBT via a two-stage framework. In the first stage, FCP trains $K$ individual policy pairs $\{(\pi_1^{(k)}, \pi_2^{(k)})\}_{k=1}^K$ by optimizing $J(\pi_1^{(k)}, \pi_2^{(k)})$ for each $k$. Each policy pair $(\pi_1^{(k)}, \pi_2^{(k)})$ may quickly converge to a distinct local optimum. Then FCP constructs a policy pool $\Pi_2 = \{\tilde{\pi}_2^{(k)}, \pi_2^{(k)}\}_{k=1}^K$ with two past versions of each converged SP policy $\pi_2^{(k)}$, denoted by $\tilde{\pi}_2^{(k)}$. In the second stage, FCP constructs a human proxy distribution $\hat{P}_H$ by randomly sampling from $\Pi_2$ and trains $\pi_A$ by optimizing $\mathbb{E}_{\hat{\pi}_H \sim \hat{P}_H}[J(\pi_A, \hat{\pi}_H)]$. We remark that, for a better cooperation, the adaptive policy $\pi_A$ should condition on the state-action history in an episode to infer the intention of its partner. Individual SP policies ensure $\hat{P}_H$ contains diverse conventions while using past versions enables $\hat{P}_H$ to cover different skill levels. So, the final policy $\pi_A$ can be forced to adapt to humans with unknown conventions or sub-optimalities. Maximum Entropy Population-based Training (MEP) (Zhao et al., 2021) is the latest variant of FCP, which adopts the population entropy as a diversity bonus in the first stage to improve the generalization of the learned $\pi_A$.

## 4 COOPERATING WITH HUMANS IN *Overcooked*: A MOTIVATING EXAMPLE

**Overcooked Game:** *Overcooked* (Carroll et al., 2019) is a fully observable two-player cooperative game developed as a testbed for human-AI cooperation. In Overcooked, players cooperatively accomplish different soup orders and serve the soups for rewards. Basic game items include onions, tomatoes, and dishes. An agent can move in the game or "interact" to trigger some events, such as grabbing/putting an item, serving soup, etc., depending on the game state. To finish an order, players

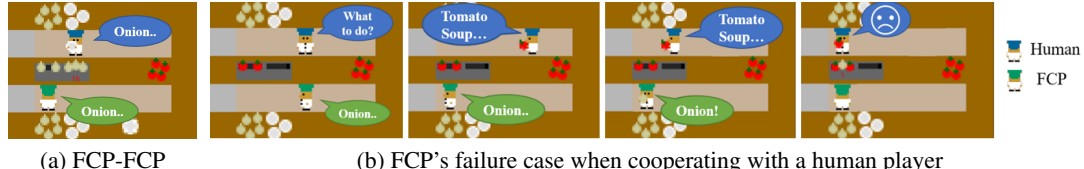

Figure 1: **Layouts in Overcooked.** From left to right are *Asymmetric Advantages*, *Coordination Ring*, *Counter Circuit*, *Distant Tomato* and *Many Orders* respectively, with orders shown below.

should put a proper amount of ingredients into the pot and cook for some time. Once a soup is finished, players should pick up the soup with a dish and serve it to get a reward. Different orders have different cooking times and different rewards. Fig. 1 demonstrates five layouts we consider, where the first three onion-only layouts are adopted from (Carroll et al., 2019), while the latter two, *Distant Tomato* and *Many Orders*, are newly introduced to include tomato orders to make the problem more challenging: an AI needs to carefully adapt its behavior to either cook onions or tomatoes according to the other player's actions.

**A Concrete Example of Human Preference:** Fig. 2 illustrates a motivating example in *Distant Tomato* (the 4th layout in Fig. 1). There are two orders: one requires three onions, and the other requires three tomatoes. We run FCP on this multi-order scenario, and all the policies in the FCP policy pool converge to the specific pattern of only cooking onion soup (Fig. 2a). Hence, the final adaptive policy by FCP only learns to grab onions and cook onion soups. Cooking tomato soup is a sub-optimal strategy that requires many extra moves, so the onion-only policy pool is exactly the solution to the FCP self-play objective under the environment reward. However, it is particularly reasonable for a human to dislike onions and accordingly only grab tomatoes in a game. To be an assistive AI, the policy should adapt its strategy to follow the human preference for tomatoes. On the contrary, as shown in Fig. 2b, the FCP policy completely ignores human moves for tomatoes and even results in poor cooperation by producing valueless wrong orders of mixed onions and tomatoes. Thus, to make an FCP agent human-assistive, the first-stage policy pool should not only contain optimal strategies (i.e., onion soups) of different conventions but also cover diverse human preferences (e.g., tomatoes) even if these preferences are sub-optimal under the environment reward.

(a) FCP-FCP    (b) FCP's failure case when cooperating with a human player

Figure 2: Motivating example. (a) FCP converges to the optimal onion soup strategy. (b) A failure case of FCP with a human partner: FCP agent corrupts the human's plan of cooking tomato soups.

## 5 METHODOLOGY

We introduce a general formulation to model human preferences and develop a tractable learning objective (Sec. 5.2). The algorithm, *Hidden-Utility Self-Play* (HSP), is summarized in Sec. 5.3.

### 5.1 HIDDEN-UTILITY MARKOV GAME

The key insight from Sec. 4 is that humans may not truthfully behave under the environment reward. Instead, humans are biased and driven by their own utility functions, which are formulated below.

**Definition:** A two-player *hidden utility Markov game* is defined as $\langle \mathcal{S}, \mathcal{A}, P, R_w, R_t \rangle$. $\langle \mathcal{S}, \mathcal{A}, P, R_t \rangle$ corresponds to the original game MDP with $R_t$ being the task reward function. $R_w$ denotes an *additional* hidden reward function. There are two players, $\pi_a$, whose goal is to maximize the task reward $R_t$, and $\pi_w$, whose goal is to maximize the hidden reward $R_w$. $R_w$ is only visible to $\pi_w$.

Let $J(\pi_1, \pi_2|R)$ denote the expected return under reward $R$ with a policy $\pi_1$ and $\pi_2$. During self-play, $\pi_a$ optimizes $J(\pi_a, \pi_w|R_t)$ while $\pi_w$ optimizes $J(\pi_a, \pi_w|R_w)$. A solution policy profile $(\pi_a^*, \pi_w^*)$ to the hidden utility Markov game is now defined by a Nash equilibrium (NE): $J(\pi_a^*, \pi_w^*|R_w) \geq J(\pi_a^*, \pi_w'|R_w), \forall \pi_w'$ and $J(\pi_a^*, \pi_w^*|R_t) \geq J(\pi_a', \pi_w^*|R_t), \forall \pi_a'$.          □

Intuitively, with a suitable hidden reward function $R_w$, we can obtain any possible (non-adaptive and consistent) human policy by solving the hidden-utility game induced by $R_w$.

**Lemma 5.1.** *Given an MDP* $M = \langle \mathcal{S}, \mathcal{A}, P, R_t \rangle$, *for any policy* $\pi : \mathcal{S} \times \mathcal{A} \to [0, 1]$, *there exists a hidden reward function* $R_w$ *such that the two-player hidden utility Markov game* $M' = \langle \mathcal{S}, \mathcal{A}, P, R_w, R_t \rangle$ *has a Nash equilibrium* $(\pi_a^*, \pi_w^*)$ *where* $\pi_w^* = \pi$.

Lemma 5.1 connects any human behavior to a hidden reward function. Then the objective of the adaptive policy $\pi_A$ in Eq. (3) can be formulated under the hidden reward function space $\mathcal{R}$ as follows.

**Theorem 5.1.** *For any $\epsilon > 0$, there exists a mapping $\tilde{\pi}_w$ where $\tilde{\pi}_w(R_w)$ denotes the derived policy $\pi_w^*$ in the NE of the hidden utility Markov game $M_w = \langle \mathcal{S}, \mathcal{A}, P, R_w, R_t \rangle$ induced by $R_w$, and a distribution $P_R : \mathcal{R} \to [0, 1]$ over the hidden reward space $\mathcal{R}$, such that, for any adaptive policy $\pi_A \in \arg\max_{\pi'} \mathbb{E}_{R_w \sim P_R}[J(\pi', \tilde{\pi}_w(R_w))]$, $\pi_A$ approximately maximizes the ground-truth objective with at most an $\epsilon$ gap, i.e., $\mathbb{E}_{\pi_H \sim P_H}[J(\pi_A, \pi_H)] \geq \max_{\pi'} \mathbb{E}_{\pi_H \sim P_H}[J(\pi', \pi_H)] - \epsilon$.*

Theorem 5.1 indicates that it is possible to derive diverse human behaviors by properly designing a hidden reward distribution $\hat{P}_R$, which can have a much lower intrinsic dimension than the policy distribution. In *Overcooked*, human preferences can be typically described by a few features, such as interaction with objects or certain type of game events, like finishing an order or delivering a soup. By properly approximating the hidden reward distribution as $\hat{P}_R$, the learning objective becomes,

$$\pi_A = \arg\max_{\pi'} \mathbb{E}_{R_w \sim \hat{P}_R}[J(\pi', \tilde{\pi}_w(R_w))] \tag{1}$$

Eq. (1) naturally suggests a two-staged solution by first constructing a policy pool $\{\tilde{\pi}_w(R) : R \sim \hat{P}_R\}$ from $\hat{P}_R$ and then training $\pi_A$ to maximize the game reward w.r.t. the induced pool.

## 5.2 Construct a Policy Pool of Diverse Preferences

**Event-based Reward Function Space:** The fundamental question is how to design a proper hidden reward function space $\mathcal{R}$. In general, a valid reward space is intractably large. Inspired by the fact that human preferences are often event-centric, we formulate $\mathcal{R}$ as linear functions over event features, namely $\mathcal{R} = \{R_w : R_w(s, a_1, a_2) = \phi(s, a_1, a_2)^T w, \|w\|_\infty \leq C_{\max}\}$. $C_{\max}$ is a bound on the feature weight $w$ while $\phi : \mathcal{S} \times \mathcal{A} \times \mathcal{A} \to \mathbb{R}^m$ specifies occurrences of different game events when taking joint action $(a_1, a_2)$ at state $s$.

**Derive a Diverse Set of Biased Policies:** We simply perform a random search over the feature weight $w$ to derive a set of diverse behaviors. We first draw $N$ samples $\{w^{(i)}\}_{i \in [N]}$ for the feature weight $w$ where $w_j^{(i)}$ is sampled uniformly from a set of values $C_j$, leading to a set of hidden reward functions $\{R_w^{(i)} : R_w^{(i)}(s, a_1, a_2) = \phi(s, a_1, a_2)^T w^{(i)}\}_{i \in [N]}$. For each hidden reward function $R_w^{(i)}$, we find an approximated NE, $\pi_w^{(i)}, \pi_a^{(i)}$, of the hidden utility Markov game induced by $R_w^{(i)}$ through self-play. The above process produces a policy pool $\{\pi_w^{(i)}\}_{i \in [N]}$ that can cover a wide range of behavior preferences.

**Policy Filtering:** We notice that the derived pool often contains a lot of similar policies. This is because the same policy can be optimal under a set of reward functions, which is typical in multi-objective optimization (Chugh et al., 2019; Tabatabaei et al., 2015). Duplicated policies simply slow down training without any help to learn $\pi_A$. For more efficient training, we adopt a behavior metric, i.e., *event-based diversity*, to only keep

---

**Algorithm 1:** Greedy Policy Selection

$S \leftarrow \{i_0\}$ where $i_0 \sim [N]$;
**for** $i = 1 \to K - 1$ **do**
  $\quad k' \leftarrow \arg\max_{k' \notin S} \mathrm{ED}(S \cup \{k'\})$;
  $\quad S \leftarrow S \cup \{k'\}$;
**end**

---

distinct ones from the initial pool. For each biased policy $\pi_w^{(i)}$, let $\mathrm{EC}^{(i)}$ denote the expected event count, i.e. $\mathbb{E}[\sum_{t=1}^T \phi(s_t, a_t) | \pi_w^{(i)}, \pi_a^{(i)}]$. We define event-based diversity for a subset $S \subseteq [N]$ by normalized pairwise EC differences, i.e., $\mathrm{ED}(S) = \sum_{i,j \in S} \sum_k c_k \cdot |\mathrm{EC}_k^{(i)} - \mathrm{EC}_k^{(j)}|$, where $c_k$ is a frequency normalization constant. Finding a subset $S^*$ of size $K$ with the optimal ED can be expensive. We simply adopt a greedy method in Algo. 1 to select policies incrementally.

## 5.3 Hidden-Utility Self-Play

Given the filtered policy pool, we train the final adaptive policy $\pi_A$ over rollout games by $\pi_A$ and randomly sampled policies from the pool, which completes our overall algorithm HSP in Algo. 2.

We implement HSP using MAPPO (Yu et al., 2021) as the RL algorithm. In the first stage, we use MLP policies for fast SP convergence. In practice, we use

---

**Algorithm 2:** Hidden-Utility Self-Play

**for** $i = 1 \to N$ **do**
  $\quad$ Train $\pi_w^{(i)}$ and $\pi_a^{(i)}$ under sampled $R_w^{(i)}$;
**end**
Run Algo. 1 to only keep $K$ policies;
Initial policy $\pi_A$;
**repeat**
  $\quad$ Rollout with $\pi_A$ and sampled $\pi_w^{(i)}$;
  $\quad$ Update $\pi_A$;
**until** *enough iterations*;

---

half of the policy pool to train biased policies and the other half to train MEP policies (Zhao et al., 2021) under the game reward. This increases the overall pool towards the game reward, leading to improved empirical performances. For the final adaptive training, as suggested in (Tang et al., 2020), we add the identity of each biased policy as an additional feature to the critic. For event-based features for the reward space, we consider event types, including interactions with basic items and events causing non-zero rewards in *Overcooked*. Full implementation details can be found in Appendix D and E.

## 6 EXPERIMENTS

**Baselines.** We compare HSP with other SP-based baselines, including Fictitious Co-Play (FCP), Maximum Entropy Population-based training (MEP), and Trajectory Diversity-based PBT (Traj-Div). All methods follow a two-stage framework with a final pool size of 36, which we empirically verified to be sufficiently large to avoid performance degradation for all methods. More analysis on pool size can be found in Appendix F.2.1. The implementation details of baselines can be found in Appendix D.2. Each policy is trained for 100M timesteps for convergence over 5 random seeds. Full training details with hyper-parameter settings can be found in Appendix E.1.

**Evaluation.** We aim to examine whether HSP can cooperate well with (1) *learned human models*, (2) *scripted* policies with strong preferences, and (3) *real humans*. We use both game reward and human feedback as evaluation metrics. We remark that since a biased human player may play a sub-optimal strategy, the game reward may not fully reflect the performance gap between the baselines and HSP. Our goal is to ensure the learned policy is effective for biased partners/humans. Therefore, we consider human feedback as the fundamental metric. Ablation studies are also performed to investigate the impact of our design choices in HSP. In tables, maximum returns or comparable returns within a threshold of 5 are marked in bold. Full results can be found in Appendix F.

### 6.1 COOPERATION WITH LEARNED HUMAN MODELS IN ONION-ONLY LAYOUTS

For evaluation with learned human models, we adopted the models provided by (Carroll et al., 2019), which *only support onion-only layouts*, including Asymm. Adv., Coord. Ring and Counter Circ.. The results are shown in Tab. 1. For a fair comparison, we reimplement all the baselines, labeled *MEP, FCP, and TrajDiv*, with the same training steps and policy pool size as HSP. We additionally take the best performance ever reported in the existing literature, labeled *Existing SOTA* in Tab. 1. Our implementation achieves substantially higher scores than Existing SOTA when evaluated with the same human proxy models. HSP further outperforms other reimplementations in Asymm. Adv. and is comparable with the best baseline in the rest. Full results of the evaluation with learned human models can be found in Appendix F.1. We emphasize that the improvement is marginal because the learned human models have limited representation power to imitate natural human behaviors, which typically cover many behavior modalities. Fig.8 in Appendix F.1.1 shows trajectories induced by the learned human models only cover a narrow subspace of trajectories played by human players. Further analysis of the learned human models can be found in Appendix F.1.1. Furthermore, our implementation of baselines achieves substantially better results than the original papers (Carroll et al., 2019; Zhao et al., 2021), which also makes the improvement margin smaller.

| | Pos. | Asymm. Adv. | Coord. Ring | Counter Circ. |
|---|---|---|---|---|
| Existing SOTA | 1 | $141.1_{(12.5)}$ | $92.7_{(7.4)}$ | $54.5_{(2.3)}$ |
| | 2 | $84.6_{(16.3)}$ | $107.3_{(6.4)}$ | $55.8_{(3.6)}$ |
| FCP | 1 | $282.8_{(9.4)}$ | $\mathbf{161.3}_{(1.6)}$ | $95.9_{(2.0)}$ |
| | 2 | $203.8_{(8.2)}$ | $\mathbf{161.0}_{(2.7)}$ | $92.7_{(1.3)}$ |
| MEP | 1 | $291.7_{(4.6)}$ | $\mathbf{161.8}_{(0.7)}$ | $\mathbf{108.8}_{(4.2)}$ |
| | 2 | $203.4_{(2.0)}$ | $\mathbf{164.2}_{(2.1)}$ | $\mathbf{111.1}_{(0.7)}$ |
| TrajDiv | 1 | $289.3_{(8.8)}$ | $150.8_{(3.1)}$ | $60.1_{(5.0)}$ |
| | 2 | $194.2_{(0.7)}$ | $142.1_{(2.3)}$ | $53.7_{(12.4)}$ |
| HSP | 1 | $\mathbf{300.3}_{(2.2)}$ | $\mathbf{160.0}_{(2.6)}$ | $\mathbf{107.4}_{(3.5)}$ |
| | 2 | $\mathbf{217.1}_{(3.3)}$ | $\mathbf{160.6}_{(3.3)}$ | $\mathbf{106.6}_{(3.0)}$ |

Table 1: Comparison of average episode reward and standard deviation when cooperating with learned human models. The Pos. column indicates the roles played by AI policies. *Existing SOTA* is the best performance ever reported in the existing literature. HSP achieves substantially higher scores than Existing SOTA. And HSP further outperforms other methods in Asymm. Adv. and is comparable with the best baseline in the rest.

### 6.2 ABLATION STUDIES

We investigate the impact of our design choices, including the construction of the final policy pool and the batch size for training the adaptive policy.

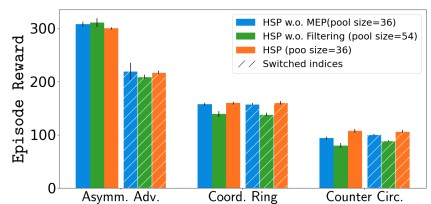

**Policy Pool Construction:** HSP has two techniques for the policy pool, i.e., (1) policy filtering to remove duplicated biased policies and (2) the use of MEP policies under the game reward for half of the pool size. We measure the performance with human proxies by turning these options off. For "*HSP w.o. Filtering*", we keep all policies by random search in the policy pool, resulting in a larger pool size of 54 (18 MEP policies and a total of 36 random search ones). For "*HSP w.o. MEP*", we exclude MEP policies from the policy pool and keep all biased policies

Figure 3: Performance of different pool construction strategies. Results suggest that it is beneficial to incorporate MEP policies and filter duplicated policies.

without filtering, which leads to the same pool size of 36. The results are shown in Fig. 3 and the detailed numbers can be found in Appendix F.2.2. By excluding MEP policies, the HSP variant (*HSP w.o. MEP*) performs worse in the more complicated layout Counter Circ. while remaining comparable in the other two simpler ones. So we suggest including a few MEP policies when possible. With policy filtering turned off, even though the policy pool size grows, the performance significantly decays in both Coord. Ring and Counter Circ. layouts, suggests that duplicated biased policies can hurt policy generalization.

**Batch Size:** We measure the training curves of the final adaptive policy under the game reward using different numbers of parallel rollout threads in MAPPO. More parallel threads indicate a larger batch size. The results in all five layouts are reported in Fig. 4. In general, we observe that a larger batch size often leads to better training performance. In particular, when the batch size is small, i.e., using 50 or 100 parallel threads, training becomes significantly unstable and even breaks in three layouts. Note that the biased policies in the HSP policy pool have particularly diverse behaviors, which cause a high policy gradient variance when training the final adaptive policy. Therefore, a sufficiently large training batch size can be critical to stable optimization. We adopt 300 parallel threads in all our experiments for a fair comparison.

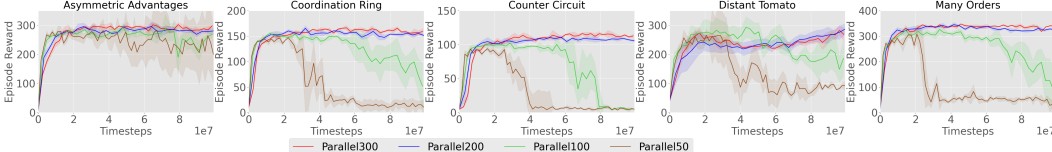

Figure 4: Average game reward by using different numbers of parallel rollout threads in MAPPO to train the final adaptive policy. More parallel threads imply a larger training batch size.

**Practical Remark:** Overall, we suggest using a pool size of 36 and including a few MEP policies for the best empirical performance. Besides, a sufficiently large training batch size can help stable optimization, and we use the same batch size for all methods for a fair comparison.

### 6.3 COOPERATION WITH SCRIPTED POLICIES WITH STRONG BEHAVIOR PREFERENCES

We empirically notice that human models learned by imitating the entire human trajectories cannot well capture a wide range of behavior modalities. So, we manually designed a set of script policies to encode some particular human preferences: *Onion/Tomato Placement*, which continuously places onions or tomatoes into the pot, *Onion/Dish Everywhere*, which keeps putting onions or dishes on the counters, *Tomato/Onion Placement and Delivery*, which puts tomatoes/onions into the pot in half of the time and tries to deliver soup in the other half of the time. For a fair comparison, we ensure that *all scripted policies are strictly different from the HSP policy pool*. More details about scripted policies and a full evaluation can be found in Appendix D.3.

We remark that scripted policies are only used for evaluation but not for training HSP. Tab. 2 shows the average game reward of all the methods when paired with scripted policies, where HSP significantly outperforms all baselines. In particular, in *Distant Tomato*, when cooperating with a strong tomato preference policy (Tomato Placement), HSP achieves a $10\times$ higher score than other baselines, suggesting that the tomato-preferred behavior is well captured by HSP.

### 6.4 COOPERATION WITH HUMAN PARTICIPANTS

We recruited 60 volunteers (28.6% female, 71.4% male, age between 18–30) by posting the experiment advertisement on a public platform and divided them into 5 groups for 5 layouts. They are provided with a detailed introduction to the basic gameplay and the experiment process. Vol-

| | Scripts | FCP | MEP | TrajDiv | HSP |
|---|---|---|---|---|---|
| Asymm. Adv. | Onion Placement | 334.8(13.0) | 330.5(14.2) | 323.6(17.0) | **376.8**(9.9) |
| | Onion Place.&Delivery | **297.7**(3.4) | **298.5**(3.4) | **290.0**(4.7) | **300.1**(4.1) |
| Coord. Ring | Onion Everywhere | 109.1(7.9) | **124.0**(3.4) | 116.9(8.9) | **121.2**(12.6) |
| | Dish Everywhere | 94.4(3.8) | 100.2(5.3) | 107.3(5.3) | **115.4**(7.4) |
| Counter Circ. | Onion Everywhere | 63.7(9.2) | 88.9(5.1) | 82.0(12.8) | **107.5**(3.5) |
| | Dish Everywhere | 57.0(5.3) | 53.0(1.8) | 57.2(2.2) | **78.5**(4.1) |
| Distant Tomato | Tomato Placement | 15.6(5.2) | 20.1(10.6) | 23.3(9.5) | **277.9**(14.3) |
| | Tomato Place.&Delivery | 177.9(6.1) | 180.4(8.7) | 164.8(19.6) | **234.6**(15.1) |
| Many Orders | Tomato Placement | 282.6(16.2) | 225.8(60.8) | 259.2(7.9) | **317.8**(9.3) |
| | Tomato Place.&Delivery | **329.1**(5.3) | **328.1**(12.6) | 295.7(2.4) | **324.5**(3.9) |

Table 2: Average episode reward and standard deviation with unseen testing scripted policies. HSP significantly outperforms all baselines.

unteers are fully aware of all their rights and experiments are approved with the permission of the department. A detailed description of the human study can be found in Appendix F.4. The experiment has two stages:

- **Warm-up Stage:** Participants could play the game freely to explore possible AI behaviors. They are asked to rank AI policies according to the degree of assistance during free plays.
- **Exploitation Stage:** Participants are instructed to achieve a score as high as possible.

We note that our user study design differs from that of the original Overcooked paper (Carroll et al., 2019). The additional warm-up stage allows for diverse human behaviors under any possible preference, suggesting a strong testbed for human-assistive AIs.

### 6.4.1 RESULTS OF THE WARM-UP STAGE

The warm-up stage is designed to test the performance of AI policies in the face of diverse human preferences. Fig. 5 visualizes the human preference for different methods reported in the warm-up stage. The unit represents the difference between the percentage of human players who prefer row partners over column partners and human players who prefer column partners over row partners. The detailed calculation method can be found in Appendix F.4.3. HSP is preferred by humans with a clear margin. Since humans can freely explore any possible behavior, the results in Fig. 5 imply the strong generalization capability of HSP. We also summarize feedback from human participants in Appendix F.4.2.

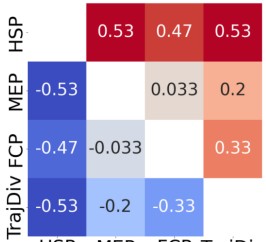

Figure 5: Human preference in the warm-up stage. The unit denotes the difference between the percentage of human players who prefer row partners over column partners and human players who prefer column partners over row partners. HSP is consistently preferred by human participants with a clear margin.

### 6.4.2 RESULTS OF THE EXPLOITATION STAGE

The exploitation stage is designed to test the scoring capability of different AIs. Note that it is possible that a human player simply adapts to the AI strategy when instructed to have high scores. So, in addition to final rewards, we also examine the emergent human-AI behaviors to measure the human-AI cooperation level. The experiment layouts can be classified into two categories according to whether the layout allows diverse behavior modes. The first category contains simple *onion-only* layouts that are taken from (Carroll et al., 2019), including Asymm. Adv., Coord. Ring and Counter Circ.. The second category contains newly introduced layouts with both onions and tomatoes, Distant Tomato and Many Orders, which allow for a much wider range of behavior modes.

**Onion-only Layouts**: Fig. 6a shows the average reward in onion-only layouts for different methods when paired with humans. Among these onion-only layouts, all methods have comparable episode reward in simpler ones (Asymm. Adv. and Coord. Ring), while HSP is significantly better in the most complex Counter Circ. layout. Fig. 6b shows the frequency of successful onion passing between the human player and the AI player. The learned HSP policy is able to use the middle counter for passing onions, while the baseline policies are less capable of this strategy.

**Layouts with Both Onions and Tomatoes:** The results and behavior analysis in Distant Tomato and Many Orders are shown as follows,

- **Distant Tomato:** In Distant Tomato, the optimal strategy is always cooking onion soups, while it is suboptimal to cook tomato soups due to the much more time spent on moving. Interestingly, our human-AI experiments found that humans may have diverse biases over onions and tomatoes. However, all learned baseline policies tend to have a strong bias towards onions and often place onions into a pot with tomatoes in it already. Tab. 3 reports the average number of such *Wrong*

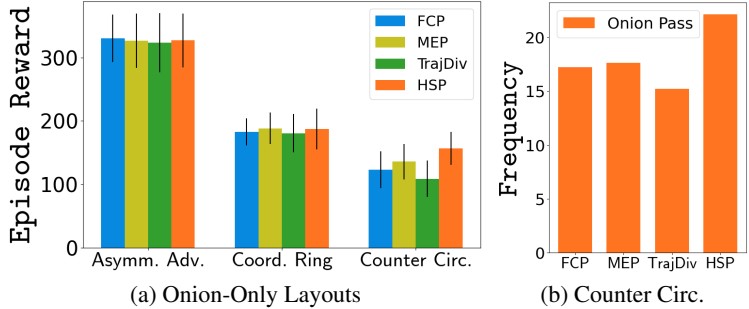

Figure 6: (a) Average episode reward in onion-only layouts of different methods when paired with humans in the exploitation stage. HSP has comparable performance with the baselines in Asymm. Adv. and Coord. Ring, and is significantly better in the most complex Counter Circ. layout. (b) The onion passing frequency in Counter Circ. shows that HSP is the most capable, among other baselines, of passing onions via the counter, suggesting better capabilities to assist humans.

    *Placements* made by different AI players. HSP makes the lowest number of wrong placements and is the only method that can correctly place additional tomatoes into a pot partially filled with tomatoes, labeled *Correct Placements*. This suggests that HSP is the only effective method to cooperate with biased human strategies, e.g., preferring tomatoes. In addition, as shown in Tab. 3, even when humans play the optimal strategy of cooking onion soups, HSP still achieves comparable performance with other methods.

- **Many Orders:** In Many Orders, an effective strategy is to utilize all three pots to cook soups. Our experiments found that baseline policies tend to ignore the middle pot. Tab. 4 shows the average number of soups picked up from the middle pot by different AI players. The learned HSP policy is much more active in taking soups from the middle pot, leading to more soup deliveries. Furthermore, HSP achieves a substantially higher episode reward than other methods, as shown in Tab. 4.

|  | FCP | MEP | TrajDiv | HSP |
|---|---|---|---|---|
| Onion-Preferred Episode Reward ↑ | **343.65** | 325.08 | 334.73 | **340.3** |
| Wrong Placements ↓ | 0.37 | 0.41 | 0.38 | **0.21** |
| Correct Placements ↑ | 0.0 | 0.0 | 0.0 | **1.41** |

Table 3: Average onion-preferred episode reward and frequency of different emergent behaviors in Distant Tomato during the exploitation stage. *Onion-Preferred Episode Reward* is the average episode reward when humans prefer onions. *Wrong Placements* and *Correct Placements* are the average numbers of wrong and correct placements into a pot partially filled with tomatoes. HSP makes the lowest number of wrong placements and is the only method that can place tomatoes correctly, suggesting that HSP is effective at cooperating with biased human strategies.

|  | FCP | MEP | TrajDiv | HSP |
|---|---|---|---|---|
| Episode Reward ↑ | 316.81 | 320.61 | 323.52 | **382.52** |
| Number of Soups Picked Up from the Middle Pot ↑ | 1.93 | 2.03 | 1.33 | **5.64** |

Table 4: Average episode reward and average number of picked-up soups from the middle pot by different AI players in Many Orders during the exploitation stage. HSP achieves significantly better performance and is much more active in taking soups from the middle pot than baselines.

## 7 CONCLUSION

We developed *Hidden-Utility Self-Play* (HSP) to tackle the problem of zero-shot human-AI cooperation by explicitly modeling human biases as an additional reward function in self-play. HSP first generates a pool of diverse strategies and then trains an adaptive policy accordingly. Experiments verified that agents trained by HSP are more assistive for humans than baselines in *Overcooked*. Although our work suggests a new research direction on this fundamentally challenging problem, there are still limitations to be addressed. HSP requires domain knowledge to design a suitable set of events. There exists some work on learning reward functions rather than assuming event-based rewards (Shah et al., 2019; Zhou et al., 2021). So a future direction is to utilize learning-based methods to design rewards automatically. Another major limitation is the computation needed to obtain a diverse policy pool. Possible solutions include fast policy transfer and leveraging a prior distribution of reward functions extracted from human data (Barreto et al., 2017a). Learning and inferring the policy representations of partners could also provide further improvement. We leave these issues as our future work.

ACKNOWLEDGMENTS

This research was supported by National Natural Science Foundation of China (No.U19B2019, 62203257, M-0248), Tsinghua University Initiative Scientific Research Program, Tsinghua-Meituan Joint Institute for Digital Life, Beijing National Research Center for Information Science, Technology (BNRist), and Beijing Innovation Center for Future Chips and 2030 Innovation Megaprojects of China (Programme on New Generation Artificial Intelligence) Grant No. 2021AAA0150000.

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

# A  THEOREM PROOFS

For simplicity, we assume state space and action space in our analysis are both discrete and finite, which is exactly the case for *Overcooked*, and the rewards $r$ are bounded: $|r(s,a)| \leq R_{\max}, \forall s \in \mathcal{S}, a \in \mathcal{A}$.

**Lemma 5.1.** *Given an MDP* $M = \langle \mathcal{S}, \mathcal{A}, P, R_t \rangle$, *for any policy* $\pi_w : \mathcal{S} \times \mathcal{A} \to [0,1]$, *there exists a hidden reward function* $R_w$ *such that the two-player hidden utility Markov game* $M' = \langle \mathcal{S}, \mathcal{A}, P, R_w, R_t \rangle$ *has a Nash equilibrium* $(\pi_a^*, \pi_w^*)$ *where* $\pi_w^* = \pi_w$.

*Proof.* Our analysis is based on the maximum entropy reinforcement learning framework (Haarnoja et al., 2018; Ziebart et al., 2008; Wulfmeier et al., 2015). Given a reward function $R$ and policies of the two players $\pi_1$ and $\pi_2$, we consider following maximum entropy RL objective for policy $\pi_i (1 \leq i \leq 2)$,

$$J_i(\pi_1, \pi_2 | R) = \mathbb{E}_\tau \left[ \sum_t \gamma^t (R(s_t, a_t^{(1)}, a_t^{(2)}) + \alpha \mathcal{H}(\pi_i(\cdot | s_t))) \Big| a_t^{(i)} \sim \pi_i(\cdot | s_t) \right]$$

We shall first constructs $\pi_a$ given policy $\pi_w$ to satisfy $J_2(\pi_w, \pi_a | R_t) \geq J_2(\pi_w, \pi_a' | R_t), \forall \pi_a'$ and secondly constructs $R_w$ such that $J_1(\pi_w, \pi_a | R_w) \geq J_1(\pi_w', \pi_a | R_w), \forall \pi_w'$ is satisfied.

**Step 1:** Construct $\pi_a$ given $\pi_w$.

Given $\pi_w$, let $\pi_a \in \arg\max_\pi J_2(\pi_w, \pi | R_t)$.

**Step 2:** Construct $R_w$ such that $J_1(\pi_w, \pi_a | R_w) \geq J_1(\pi_w', \pi_a | R_w), \forall \pi_w'$ is satisfied given $\pi_w$ and $\pi_a$.

Given a fixed partner $\pi_a$, by regarding $\pi_a$ as part of the environment dynamics, we could consider the dynamics for $\pi_w$ in a single-agent MDP $M' = \langle \mathcal{S}, \mathcal{A}, P', R_w, \gamma \rangle$ where $\mathcal{S}$ is the state space, $\mathcal{A}$ is the action space, $P'$ denotes the transition probability and $R_w$ is the reward function to construct. More specifically, $P'$ is defined as,

$$P'(s'|s,a) = \sum_{\tilde{a}} P(s'|s, a, \tilde{a}) \cdot \pi_a(\tilde{a}|s) \tag{2}$$

In $M'$, given reward $R_w$, the objective of $\pi_w$ becomes,

$$\max_\pi \mathbb{E}_\tau \left[ \sum_t \gamma^t (R_w(s_t, a_t) + \alpha \mathcal{H}(\pi(s_t))) \Big| a_t \sim \pi(s_t) \right] \tag{3}$$

The value function and the Q function could be defined as,

$$V(s) = \mathbb{E}_\tau \left[ \sum_t \gamma^t (R_w(s_t, a_t) + \alpha \mathcal{H}(\pi_w(s_t))) \Big| a_t \sim \pi_w(s_t), s_0 = s \right] \tag{4}$$

$$= \sum_a \pi_w(a|s)(R_w(s,a) + \gamma \mathbb{E}_{s'}[V(s')|s,a]) + \alpha \mathcal{H}(\pi_w(s)) \tag{5}$$

$$Q(s,a) = R_w(s,a) + \gamma \cdot \mathbb{E}_{s'}[V(s')|s,a] \tag{6}$$

It is sufficient to construct $R_w$ such that $V(s)$ is a stable point of the *Bellman backup operator* (Sutton & Barto, 2018) $\mathcal{T}^*$ under some $R_w$:

$$(\mathcal{T}^* V)(s) = \max_{d: \sum_a d(a) = 1} \alpha \mathcal{H}(d) + \sum_a d(a)(R_w(s,a) + \gamma \mathbb{E}_{s'}[V(s')|s,a]) \tag{7}$$

Now we assume $V(s)$ is a stable point for Eq. 7 and construct $R_w$. For all $s \in \mathcal{S}$, $\pi_w(\cdot|s)$ should be a solution to the following maximization problem,

$$\max_d \alpha \mathcal{H}(d) + \sum_a d(a)Q(s,a) \tag{8}$$

$$s.t. \quad \sum_a d(a) = 1 \tag{9}$$

Applying KKT conditions over the above optimization problem indicates that,

$$\pi_w(\cdot|s) \propto \exp(Q(s,\cdot)/\alpha), \forall s \tag{10}$$

Let $\pi_w^*(s) = \arg\max_a \pi_w(a|s), V^*(s) = \max_a Q(s,a), A(s,a) = Q(s,a) - V^*(s)$. By Eq. 10, we also have

$$A(s,a) = \alpha(\log \pi_w(a|s) - \log \pi_w(\pi_w^*(s)|s)) \tag{11}$$

By definition of value function $V(s)$,

$$V(s) = \sum_a \pi_w(a|s)Q(s,a) + \alpha\mathcal{H}(\pi_w(s)) \tag{12}$$

$$= \sum_a \pi_w(a|s)(A(s,a) + V^*(s)) + \alpha\mathcal{H}(\pi_w(s)) \tag{13}$$

$$= \sum_a \pi_w(a|s)A(s,a) + V^*(s) + \alpha\mathcal{H}(\pi_w(s)) \tag{14}$$

$$= \sum_a \pi_w(a|s)A(s,a) + R_w(s,\pi_w^*(s)) + \gamma\mathbb{E}_{s'}[V(s')|s' \sim P'(s,\pi_w^*(s))] + \alpha\mathcal{H}(\pi_w(s)) \tag{15}$$

$$= \mathbb{E}_\tau \left[ \sum_t \gamma^t \left( \sum_{a'} \pi_w(a'|s)A(s,a') + R_w(s_t,a_t) + \alpha\mathcal{H}(\pi_w(s_t)) \right) \Big| a_t = \pi_w^*(s_t) \right] \tag{16}$$

Let $b(s) = R_w(s,\pi_w^*(s))$. Then $V(s)$ is determined given $\pi_w$ and $b$,

$$V(s) = \mathbb{E}_\tau \left[ \sum_t \gamma^t \left( \sum_{a'} \pi_w(a'|s)A(s,a') + b(s_t) + \alpha\mathcal{H}(\pi_w(s_t)) \right) \Big| a_t = \pi_w^*(s_t) \right] \tag{17}$$

By $A(s,a) = \alpha(\log \pi_w(a|s) - \log \pi_w(\pi_w^*(s)|s)) = Q(s,a) - V^*(s)$,

$$\alpha(\log \pi_w(a|s) - \log \pi_w(\pi_w^*(s)|s)) = R_w(s,a) + \gamma\mathbb{E}_{s'}[V(s')|s' \sim P'(s,a)] - V^*(s) \tag{18}$$

$$R_w(s,a) = \alpha \log \left( \frac{\pi_w(a|s)}{\pi_w(\pi_w^*(s)|s)} \right) - \gamma\mathbb{E}_{s'}[V(s')|s' \sim P'(s,a)] + V^*(s) \tag{19}$$

To summarize, for policy $\pi_w$, we can construct a valid hidden reward function $R_w$ via following process,

1. Choose a function $b : \mathcal{S}' \to \mathbb{R}$.

2. Compute $A(s,a)$ by Eq. 11.

3. Compute $V(s)$ and $V^*(s)$ by Eq. 17.

4. Construct $R_w(s,a)$ by $R_w(s,\pi_w^*(s)) = b(s)$ and Eq. 19.

$\square$

Now we show that, for any $b : \mathcal{S}' \to \mathbb{R}$, under $R_w$ constructed by the above process, $V(s)$ is a stable point of the Bellman backup operator $\mathcal{T}^*$. This is straightforward. First, constructed $R_w$ ensures

that $\alpha(\log \pi_w(a|s) - \log \pi_w(\pi_w^*(s)|s)) = Q(s,a) - V^*(s)$ (Eq. 19) and therefore $\pi_w(a|w) \propto \exp(Q(s,a)/\alpha)$, which means $\pi_w$ is a solution for the maximization problem 8. So

$$(\mathcal{T}^*V)(s) = \max_d \alpha\mathcal{H}(d) + \sum_a d(a)(R_w(s,a) + \gamma\mathbb{E}_{s'}[V(s')]) \tag{20}$$

$$= \sum_a \pi_w(a|s)Q(s,a) + \alpha\mathcal{H}(\pi_w(s)) = V(s). \tag{21}$$

**Theorem 5.1.** *For any $\epsilon > 0$, there exists a mapping $\tilde{\pi}_w$ where $\tilde{\pi}_w(R_w)$ denotes the derived policy $\pi_w^*$ in the NE of the hidden utility Markov game $M_w = \langle \mathcal{S}, \mathcal{A}, P, R_w, R_t \rangle$ induced by $R_w$, and a distribution $P_R : \mathcal{R} \to [0,1]$ over the hidden reward space $\mathcal{R}$, such that, for any adaptive policy $\pi_A \in \arg\max_{\pi'} \mathbb{E}_{R_w \sim P_R}[J(\pi', \tilde{\pi}_w(R_w))]$, $\pi_A$ approximately maximizes the ground-truth objective with at most an $\epsilon$ gap, i.e., $\mathbb{E}_{\pi_H \sim P_H}[J(\pi_A, \pi_H)] \geq \max_{\pi'} \mathbb{E}_{\pi_H \sim P_H}[J(\pi', \pi_H)] - \epsilon$.*

*Proof.* Let $K(K > |\mathcal{A}|)$ be a large positive integer. We construct a discretization of the policy space $\Pi$ by $\Pi_K = \{\pi : \pi(a|s) = \frac{i}{K}$ where $i \in [K], \forall s \in \mathcal{S}, a \in \mathcal{A}$ and $\sum_a \pi(a|s) = 1, \forall s \in \mathcal{S}\}$. Note that $\Pi_K$ is finite, i.e. $|\Pi_K| \leq (K+1)^{|\mathcal{S}| \cdot |\mathcal{A}|}$. Let $M = |\Pi_K|$ and $\pi_1, \pi_2, \cdots, \pi_M$ be an ordering of the policies in $\Pi_K$. For simplicity of notation, let $\delta = \frac{|\mathcal{A}|}{K}$.

Given the discretization $\Pi_K$, it's straightforward to specify the nearest policy $\hat{\pi} \in \Pi_K$ for any policy $\pi \in \Pi$. Formally, for any policy $\pi \in \Pi$, let $G(\pi) = \arg\min_{i=1,...,M} \sum_{s,a} |\pi(a|s) - \pi_i(a|s)|$. An obvious property of $G$ is that, $\forall s \in \mathcal{S}, ||\pi(\cdot|s) - G(\pi)(\cdot|s)||_\infty \leq \frac{|\mathcal{A}|}{K} = \delta$.

For two policies $\pi_1$ and $\pi_2$, consider $\pi_1$ playing with $\pi_2$ and $G(\pi_2)$ respectively. Since the action distribution of $\pi_2$ and $G(\pi_2)$ at each state differ at most $\delta$, we have follows,

$$|J(\pi_1, \pi_2) - J(\pi_1, G(\pi_2))| \leq \sum_t \gamma^t \cdot (1-\delta)^t \cdot \delta \cdot \frac{2R_{\max}}{1-\gamma} \leq \frac{2\delta R_{\max}}{(1-\gamma)^2} \tag{22}$$

We can then derive a discretized approximation of the ground-truth policy distribution $P_H$ as follows,

$$\hat{P}_H(\pi) = \Pr_{\pi' \sim P_H}[\pi = G(\pi')] \tag{23}$$

We could show that the difference between the objective under the ground-truth policy distribution $P_H$ and that under the approximated policy distribution $\hat{P}_H$ is bounded. By Eq. 22, for any adaptive policy $\pi_A$,

$$\left|\mathbb{E}_{\pi_H \sim \hat{P}_H}[J(\pi_A, \pi_H)] - \mathbb{E}_{\pi_H \sim P_H}[J(\pi_A, \pi_H)]\right| = \left|\mathbb{E}_{\pi_H \sim P_H}[J(\pi_A, G(\pi_H)) - J(\pi_A, \pi_H)]\right| \tag{24}$$

$$\leq \frac{2\delta R_{\max}}{(1-\gamma)^2} \tag{25}$$

On the other hand, consider following an iterative process to find hidden reward functions for policies in $\Pi_K$. For $i = 1..M$, we find hidden reward function $R_w^{(i)}$ where $R_w^{(i)} \notin \{R_w^{(j)} | 1 \leq j \leq i-1\}$ and $R_w^{(i)}$ could be constructed from $\pi_i$ as in Lemma 5.1. Notice that, by construction rule in Lemma 5.1, such $R_w^{(i)}$ must exists since we can specify arbitrary $b : \mathcal{S} \to \mathbb{R}$.

Let $\tilde{\pi}_w(R_w^{(i)}) = \pi_i, \forall i = 1 \ldots M$ and the hidden reward distribution $P_R$ be $P_R(R_w^{(i)}) = \hat{P}_H(\pi_i), \forall i = 1 \cdots M$. We immediately see that, for any adaptive policy $\pi_A$, the objective is equivalent under the approximated policy distribution $\hat{P}_H$ and hidden reward function distribution $P_R$,

$$\mathbb{E}_{R_w \sim P_R}[J(\pi_A, \tilde{\pi}_w(R_w))] = \mathbb{E}_{\pi_H \sim \hat{P}_H}[J(\pi_A, \pi_H)] \tag{26}$$

Finally, for any adaptive policy $\pi_A \in \arg\max_{\pi'} \mathbb{E}_{R_w \sim P_R}[J(\pi', \tilde{\pi}_w(R_w))]$ and any policy $\pi' \in \Pi$,

$$\mathbb{E}_{\pi_H \sim P_H}[J(\pi_A, \pi_H)] \geq \mathbb{E}_{\pi_H \sim \hat{P}_H}[J(\pi_A, \pi_H)] - \frac{2\delta R_{\max}}{(1-\gamma)^2} \tag{27}$$

$$= \mathbb{E}_{R_w \sim P_R}[J(\pi_A, \tilde{\pi}_w(R_w))] - \frac{2\delta R_{\max}}{(1-\gamma)^2} \tag{28}$$

$$\geq \mathbb{E}_{R_w \sim P_R}[J(\pi', \tilde{\pi}_w(R_w))] - \frac{2\delta R_{\max}}{(1-\gamma)^2} \tag{29}$$

$$= \mathbb{E}_{\pi_H \sim \hat{P}_H}[J(\pi', \pi_H)] - \frac{2\delta R_{\max}}{(1-\gamma)^2} \tag{30}$$

$$\geq \mathbb{E}_{\pi_H \sim P_H}[J(\pi', \pi_H)] - \frac{4\delta R_{\max}}{(1-\gamma)^2} \tag{31}$$

Let $K \geq \frac{4|\mathcal{A}|R_{\max}}{\epsilon(1-\gamma)^2}$ and we have $\mathbb{E}_{\pi_H \sim P_H}[J(\pi_A, \pi_H)] \geq \max_{\pi'} \mathbb{E}_{\pi_H \sim P_H}[J(\pi', \pi_H)] - \epsilon$.

$\square$

# B  ENVIRONMENT DETAILS

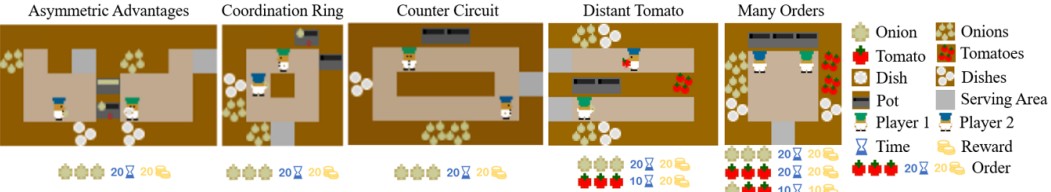

Figure 7: All 5 layouts used in our work (from left to right): *Asymmetric Advantage*, *Coordination Ring*, *Counter Circuit*, *Distant Tomato*, and *Many Orders*, each featuring specific cooperation patterns we want to study.

## B.1  DESCRIPTION

The Overcooked Environment, first introduced in (Carroll et al., 2019), is based on the popular video game Overcooked where multiple players cooperate to finish as many orders as possible within a time limit. In this simplified version of the original game, two chiefs, each controlled by a player (either human or AI), work in grid-like *layouts*. Chiefs can move between non-table tiles and interact with table tiles by picking up or placing objects. Ingredients (e.g., onions and tomatoes) and empty dishes can be picked up from the corresponding dispenser tiles and placed on empty table tiles or into the pots. The typical pipeline for completing an order is (1) players put appropriate ingredients into a pot; (2) a pot starts cooking automatically once filled and takes a certain amount of time (depending on the recipe) to finish; (3) a player harvests the cooked soup with an empty dish and deliver it to the serving area.

The **observation** for an agent includes the whole layout, items on the counter and pots, player positions, orders, and time. The possible **actions** are up, down, left, right, no-op, and "interacting" with the tile the player is facing. **Reward** is given to both agents upon successful soup delivery, with the amount varying with the type of soup. An episode of the game terminates when the time limit is reached.

The environment used in (Carroll et al., 2019) has only onions as ingredients and onion soups as orders. In our work, we evaluate all methods in three of them, namely *Asymmetric Advantage*, *Coordination Ring*, and *Counter Circuit*, each designed to enforce a specific cooperation pattern.

Our work introduces two new layouts: *Distant Tomato* and *Many orders*, with new ingredients and order types to make cooperation more challenging. In *Distant Tomato*, a dish of onion soup takes 20 ticks to finish and gives 20 rewards when delivered, while a tomato soup takes 10 ticks and gives the same reward but needs more movements to get the ingredient. The two players need to agree on which type of soup to cook in order to reach a high score. Failure in cooperation may result in tomato-onion soups that give no reward. In many orders, there are three types of orders: onion,

tomato, and 1-onion-2-tomato. To fully utilize the three pots, the players need to work seamlessly in filling not just the pots near each of them but also the pot in the middle.

We show all the layouts in Fig.7. and conclude the cooperation pattern of our interest as follows.

- *Asymmetric Advantage* tests whether the players can choose a strategy to their strengths.
- *Coordination Ring* requires the players not to block each other when traveling between the two corners.
- *Counter Circuit* embeds a non-trivial but efficient strategy of passing onions through the middle counter, which needs close cooperation.
- *Distant Tomato* and *Many Orders* both encourage the players to reach an agreement on the fly in order to achieve a high reward.

### B.2 EVENTS

In Overcooked, we consider the following events for random search in HSP and reward shaping during training of all methods:

- putting an onion/tomato/dish/soup on the counter,
- picking up an onion/tomato/dish/soup from the counter,
- picking up an onion from the onion dispenser,
- picking up a tomato from tomato dispenser,
- picking up a dish from the dish dispenser,
- picking up a ready soup from the pot with a dish,
- placing an onion/tomato into the pot,
- valid placement: after the placement, we can finish an order with a positive reward by placing other ingredients,
- optimal placement: the placement is optimal if the maximum order reward we can achieve for this particular pot is not decreased after the placement,
- catastrophic placement: the placement is catastrophic if the maximum order reward we can achieve for this particular pot decreases from positive to zero after the placement,
- useless placement: the placement is useless if the maximum order reward we can achieve for this particular pot is already zero before the placement,
- useful dish pickup: picking up a dish is useful when there are no dishes on the counter, and the number of dishes already taken by players is less than the total number of unready and ready soups,
- delivering a soup to the serving area.

Additionally, in *Distant Tomato*, we consider the following events only for reward shaping,

- placing a tomato into an empty pot,
- optimal tomato placement: the placement is optimal and a tomato placement,
- useful tomato pickup: the agent picks up a tomato when the partner isn't holding a tomato, and there is a pot that is not full but only has tomatoes in it.

## C OVERCOOKED VERSION

In our experiments, we use two versions of Overcooked for a fair comparison with prior works and introduce challenging layouts. One version, in which we tested *Asymmetric Advantages*, *Coordination Ring* and *Counter Circuit*, is consistent with the "neurips2019" branch in the released GitHub repository of (Carroll et al., 2019). We remark that MEP (Zhao et al., 2021) also follows this version. Following this also allows us to perform an evaluation with human proxy models provided in the released code of (Carroll et al., 2019). The other version is an up-to-date version of Overcooked, which supports tomatoes and user-defined orders. We notice that a pot automatically starts cooking soup once there are three items in it in the former version, while it requires an additional "interact" action to start cooking in the latter version. This additional "interact" is required in the latter version since it supports orders with different amounts of ingredients. However, having an additional "interact" significantly influences a human player's interactive experience. Therefore, we make modifications on the latter version to restrict orders to 3 items and support auto-cooking when there are 3 items. For more details, please refer to the released code.

# D  IMPLEMENTATION DETAILS

## D.1  HSP

---

**Algorithm 3:** Hidden-Utility Self-Play

---

**for** $i = 1 \rightarrow N$ **do**
  Train $\pi_w^{(i)}$ and $\pi_a^{(i)}$ under sampled $R_w^{(i)}$;
**end**
Run greedy policy selection to only keep $K$ policies;
Initial policy $\pi_A$;
**repeat**
  Rollout with $\pi_A$ and sampled $\pi_w^{(i)}$;
  Update $\pi_A$;
**until** *enough iterations*;

---

The pseudocode of HSP is shown in Algo. 3. We implemented HSP on top of MAPPO (Yu et al., 2021). Following the standard practice, we use multiprocessing to collect trajectories in parallel and then update the models. In the first stage, we use MLP policies, which empirically yield better results. In the second stage, we use RNN policies so that the adaptive policy could infer the intention of its partner by observing the history of its partner and make decisions accordingly for better adaptation. As suggested in(Tang et al., 2020), we add the identities of the policies in the policy pool as an additional feature to the critic. For better utilization of the computation resources, each environment sub-process loads a uniformly sampled policy and performs inference on CPUs, while the inference of the adaptive policy is batched across sub-processes in a GPU.

## D.2  BASELINES

For a fair comparison, we implement all baselines to be two-staged and train layout-specific agents.

We remark that our implementation of MEP achieves substantially higher scores than reported in the original paper (Zhao et al., 2021) when evaluated with the same human proxy models as MEP. All baselines are implemented with techniques stated above: loading policies from the pool per sub-process and the additional feature of identities of policies in the policy pool. We detail the baselines here and point out the difference with the original papers,

**FCP(Strouse et al., 2021):** We list the differences between our implementation and the original FCP as follows,

1. The original FCP uses image-based egocentric observations, while we use feature-based observations as provided in Overcooked.

2. The original FCP uses a pool size of 96 while we use 36. We empirically found 36 a sufficiently large pool size in our experiments. As shown in Table 21, in the three layouts that have human proxy models, there is no significant difference between using a pool size of 36 and of 72.

**MEP(Zhao et al., 2021):** We list the differences between our implementation and the original MEP as follows,

1. While the released code of MEP uses MLP policy in the second training stage, we found RNN policy to work better. Intuitively, for better cooperation, the adaptive policy should infer the intention of its partner by observing the state-action history.

2. MEP uses a pool size of 15 while we use 36.

3. MEP uses prioritized sampling in the second stage, which favors weak policies in the pool, while we adopt uniform sampling for MEP since we found prioritized sampling not helpful with our carefully tuned implementation (shown in Table 5).

4. In the released code of MEP, the policy updates are performed on data against only one policy from the pool, while we perform policy updates on data against many policies from the pool. This avoids the update from being biased towards some specific policies.

**TrajDiv(Lupu et al., 2021):** While the original TrajDiv is tested in hand-crafted MDPs and Hanabi, we test TrajDiv in Overcooked. Although (Lupu et al., 2021) suggests training the adaptive policy and the policy pool together in a single stage, we choose to follow MEP and FCP to have a two-staged design that trains the adaptive policy in the second stage.

|                      | Pos. | Asy. Adv. | Coor. Ring | Coun. Circ. |
|----------------------|------|-----------|------------|-------------|
| Uniform Sampling     | 1    | $291.7_{(4.6)}$ | $161.8_{(0.7)}$ | $108.8_{(4.2)}$ |
|                      | 2    | $203.4_{(2.0)}$ | $164.2_{(2.1)}$ | $111.1_{(0.7)}$ |
| Prioritized Sampling | 1    | $284.6_{(3.2)}$ | $161.2_{(1.4)}$ | $94.4_{(2.3)}$ |
|                      | 2    | $218.8_{(2.4)}$ | $167_{(4.5)}$ | $99.8_{(1.8)}$ |

Table 5: Average episode reward and standard deviation (over 5 seeds) with different sampling methods of MEP. The "1" and "2" indicates the roles played by AI policies.

|               | Biased | Scripted |
|---------------|--------|----------|
| Asymm. Adv.   | 0.56   | **0.85** |
| Coord. Ring   | 0.59   | **0.72** |
| Counter Circ. | 0.56   | **0.73** |
| Dist. Tomato  | 0.70   | **1.90** |
| Many Orders   | 0.55   | **1.00** |

Table 6: The average event-based difference of biased and scripted policies respectively.

### D.3 SCRIPTED POLICIES

To evaluate all methods with policies that have strong preferences, we consider the following scripted policies,

- *Onion/Tomato/Dish Everywhere* continuously tries to put onions, tomatoes or dishes over the counter.

- *Onion/Tomato Placement* always tries to put onion or tomato into the pot.

- *Delivery* delivers a ready soup to the serving area whenever possible.

- *Onion/Tomato Placement and Delivery* puts tomatoes/onions into the pot in half of the time and tries to deliver soup in the other half of the time.

For *Counter Circuit*, we additionally consider a scripted policy, named *Onion to Middle Counter*, which keeps putting onions randomly over the counter in the middle of the layout.

Input to these scripted policies is the ground-truth state of the game, which is accessible via the game simulator. When a scripted policy is unable to finish the event of its interest at some state, the scripted policy would walk to a random empty grid. For example, *Onion Placement* would choose a random walk when all pots are full. We ensure that these scripted policies are strictly different from policies in the policy pool of HSP. For more details, please refer to the released code.

We also provide evidence to show scripted policies are sufficiently different from those in the training pool. We use the expected event count of scripted and biased policies to support our claim. Recall that expected event count for a pair of policy $\pi_a, \pi_b$ is $EC(\pi_a, \pi_b) = \mathbb{E}[\sum_{t=1}^{T} \phi(s_t, a_t)|\pi_a, \pi_b]$. Let $\pi_{HSP}$ be the HSP adaptive policy, $\{\pi_w^{(n)}\}_{n \in [N]}$ be the set of biased policies in the training pool, and $\{\pi_s^{(m)}\}_{m \in M}$ be the set of scripted policies. For convenience, let $\Pi = \{\pi_w^{(n)}\}_{n \in [N]} \cup \{\pi_s^{(m)}\}_{m \in M}$ be the **union** of biased policies and scripted policies. For each policy $\pi' \in \Pi$, we measure how close it is to the rest of policies in $\Pi$ in the expected event count, i.e. the *event-based difference* $\text{EventDiff}_\Pi(\pi') = \min_{\pi'' \in \Pi \setminus \{\pi'\}} \sum_k c_k \cdot |EC_k(\pi', \pi_{HSP}) - EC_k(\pi'', \pi_{HSP})|$ where $c_k$ is a frequency normalization constant. Then a large event-based difference indicates that $\pi'$ is sufficiently different from other policies in $\Pi$. We calculate the event-based difference for all biased and scripted policies. Table. 6 reports the average event-based difference between biased and scripted policies, respectively. Scripted policies consistently have a larger average event-based difference, indicating scripted policies are sufficiently different from biased policies, which are used for training the HSP adaptive policy.

# E  TRAINING DETAILS

## E.1  HYPERPARAMETERS

HSP and baselines are all two-staged solutions by first constructing a policy pool and then training an adaptive policy $\pi_A$ to maximize the game reward w.r.t. the induced pool.

The network architecture in both two stages is composed of 3 convolution layers with max pooling. Hyperparameters of these layers are listed in Table 7. Each layer is followed by a max pooling layer with a kernel size of 2. For MLP policies, we add two linear layers after the convolution. For RNN policies, we add a 1-layer GRU after the convolution and two linear layers after the GRU layer. The hidden sizes for these linear layers and the GRU layer are all 64. We use ReLU as the activation function between layers and LayerNorm after GRU and linear layers except the last one. The output is a 6-dim vector denoting the categorical action distribution.

Common hyperparameters for all methods in 5 layouts are listed in Table 8 and Table 9. Specifically, for MEP, we use the suggested hyperparameters from the original paper (Zhao et al., 2021). Detailed hyperparameters of MEP are shown in Table 10, where population entropy coef. adjusts the importance of the population entropy term. Detailed hyperparameters of TrajDiv are shown in Table 11, where traj. gamma is the discounting factor used in local action kernel and diversity coef. adjusts the importance of the diversity term. For each one of MEP, FCP and TrajDiv, we train 12 policies in the first stage and, following the convention of MEP (Zhao et al., 2021) and FCP (Strouse et al., 2021), take the init/middle/final checkpoints for each policy to build up the policy pool, leading to a pool size of 36. For HSP, we use a random search to first train 36 biased policies and then filter out 18 biased policies from them. We then combine these biased policies and past checkpoints of 6 policies in the policy pool of MEP to build up the policy pool of HSP, again leading to a pool size of 36.

| Layer | Out Channels | Kernel Size | Stride | Padding |
|-------|--------------|-------------|--------|---------|
| 1 | 32 | 3 | 1 | 1 |
| 2 | 64 | 3 | 1 | 1 |
| 3 | 32 | 3 | 1 | 1 |

Table 7: CNN feature extractor hyperparameters.

| common hyperparameters | value |
|------------------------|-------|
| entropy coef. | 0.01 |
| gradient clip norm | 10.0 |
| GAE lambda | 0.95 |
| gamma | 0.99 |
| value loss | huber loss |
| huber delta | 10.0 |
| mini batch size | batch size / mini-batch |
| optimizer | Adam |
| optimizer epsilon | 1e-5 |
| weight decay | 0 |
| network initialization | Orthogonal |
| use reward normalization | True |
| use feature normalization | True |
| learning rate | 5e-4 |
| parallel environment threads | 100 |
| ppo epoch | 15 |
| environment steps | 10M |
| episode length | 400 |
| reward shaping horizon | 100M |

Table 8: Common hyperparameters in the first stage.

## E.2 Constructing the Policy Pool for HSP

To construct the policy pool for HSP, we perform a random search over possible hidden reward functions. Each reward function is formulated as a linear function over the event-based features, i.e. $\mathcal{R} = \{R_w : R_w(s, a_1, a_2) = \phi(s, a_1, a_2)^T w, ||w||_\infty \leq C_{\max}\}$ where $\phi : \mathcal{S} \times \mathcal{A} \times \mathcal{A} \rightarrow \mathbb{R}^m$ specifies occurrences of different events when taking joint action $(a_1, a_2)$ at state $s$. To perform random search, instead of directly sampling each $w_j$ from the section $[-C_{\max}, C_{\max}]$, we sample each $w_j$ from a set of possible values $C_j$. We detail the $C_j$ for each event on each layout here. Tab. 12 shows $C_j$ in *Asymmetric Advantages*, *Coordination Ring* and *Counter Circuit*. Tab. 13 and Tab. 14 show $C_j$ in *Distant Tomato* and *Many Orders* respectively. A detailed description of the events is shown in Sec. B.2. Note that in addition to events, we also include order reward as one element in a random search.

To filter out duplicated policies, we define an *event-based diversity* for a subset $S$, i.e. $\text{ED}(S) = \sum_{i,j \in S} \sum_k c_k \cdot |\text{EC}_k^{(i)} - \text{EC}_k^{(j)}|$ where $\text{EC}_k^i$ is the expected number of occurrences of event type $k$ for biased policy $\pi_w^{(i)}$. The coefficient $c_k$ balances the importance of different kinds of events. We simply set $c_k$ as a normalization constant, i.e. $c_k = \left(\max_{i \in [N]} \text{EC}_k^{(i)}\right)^{-1}$.

| common hyperparameters | value |
|---|---|
| entropy coef. | 0.01 |
| gradient clip norm | 10.0 |
| GAE lambda | 0.95 |
| gamma | 0.99 |
| value loss | huber loss |
| huber delta | 10.0 |
| mini batch size | batch size / mini-batch |
| optimizer | Adam |
| optimizer epsilon | 1e-5 |
| weight decay | 0 |
| network initialization | Orthogonal |
| use reward normalization | True |
| use feature normalization | True |
| learning rate | 5e-4 |
| parallel environment threads | 300 |
| ppo epoch | 15 |
| environment steps | 100M |
| episode length | 400 |
| reward shaping horizon | 100M |
| policy pool size | 36 |

Table 9: Common hyperparameters in the second stage.

| hyperparameters | value |
|---|---|
| population entropy coef. | 0.01 |

Table 10: MEP hyperparameters in the first stage.

| hyperparameters | value |
|---|---|
| traj. gamma | 0.5 |
| diversity coef. | 0.1 |

Table 11: TrajDiv hyperparameters in the first stage.

| Event | $C_j$ |
|---|---|
| Picking up an onion from onion dispenser | -10, 0, 10 |
| Picking up a dish from dish dispenser | 0, 10 |
| Picking up a ready soup from the pot | -10, 0, 10 |
| Placing an onion into the pot | -10, 0, 10 |
| Delivery | -10, 0 |
| Order reward | 0, 1 |

Table 12: $C_j$ for random search in Asymmetric Advantages, Coordination Ring and Counter Circuit.

| Event | $C_j$ |
|---|---|
| Picking up an onion from onion dispenser | -5, 0, 5 |
| Picking up a tomato from tomato dispenser | 0, 10, 20 |
| Picking up a dish from dish dispenser | 0, 10 |
| Picking up a soup | -5, 0, 5 |
| Viable placement | -10, 0, 10 |
| Optimal placement | -10, 0, 10 |
| Catastrophic placement | 0, 10 |
| Placing an onion into the pot | -10, 0, 10 |
| Placing a tomato into the pot | -10, 0, 10 |
| Delivery | -10, 0 |
| Order reward | 0, 1 |

Table 13: $C_j$ for random search in Distant Tomato.

### E.3 REWARD SHAPING

We use reward shaping during training in all layouts, detailed as follows,

- In the first stage, the reward shaping for *Asymmetric Advantages*, *Coordination Ring* and *Counter Circuit* is shown in Table. 15 and that for *Distant Tomato* and *Many Orders* is shown in Table. 17. Note that we do not use reward shaping when training biased policies for HSP in the first stage.
- In the second stage, the reward shaping for *Asymmetric Advantages*, *Coordination Ring* and *Counter Circuit* is shown in Table. 16. Reward shaping for *Many Orders* is shown in Table. 18 and that for *Distant Tomato* is shown in Table. 19. The factor of shaped reward anneals from 1 to 0 during the whole course of training in all layouts except *Distant Tomato*, in which the factor anneals from 1 to 0.5.

## F FULL RESULTS

### F.1 COOPERATION WITH LEARNED HUMAN MODELS

Table 20 shows average episode reward and standard deviation (over 5 seeds) on 3 layouts for different methods played with human proxy policies. All values within 5 standard deviations of the maximum episode return are marked in bold. These three simple layouts may not fully reflect the performance gap between the baselines and HSP. The results with learned human models are reported for a fair comparison with existing SOTA methods. Besides, our implementation of the baselines achieves substantially better results than their original papers with the same human proxy models, making the improvement margin look smaller. We also remark that the learned human models have limited representation power to imitate natural human behaviors that typically cover many behavior modalities. Here we give empirical evidence of the learned human models failing to fully reflect human behaviors.

### F.1.1 EMPIRICAL EVIDENCE

The original Overcooked paper (Carroll et al., 2019) collected human-play trajectories. We then collect game trajectories played by the learned human models and compare them with human-play

| Event | $C_j$ |
|---|---|
| Picking up an onion from onion dispenser | -5, 0, 5 |
| Picking up a tomato from tomato dispenser | 0, 10, 20 |
| Picking up a dish from dish dispenser | 0, 5 |
| Picking up a soup | -5, 0, 5 |
| Viable placement | -10, 0, 10 |
| Optimal placement | -10, 0 |
| Catastrophic placement | 0, 10 |
| Placing an onion into the pot | -3, 0, 3 |
| Placing a tomato into the pot | -3, 0, 3 |
| Delivery | -10, 0 |
| Order reward | 0, 1 |

Table 14: $C_j$ for random search in Many Orders.

| Event | Value |
|---|---|
| Optimal placement | 3 |
| Picking up a dish from dish dispenser | 3 |
| Picking up a ready soup from the pot | 5 |

Table 15: Reward shaping for Asymmetric Advantages, Coordination Ring and Counter Circuit in the first stage.

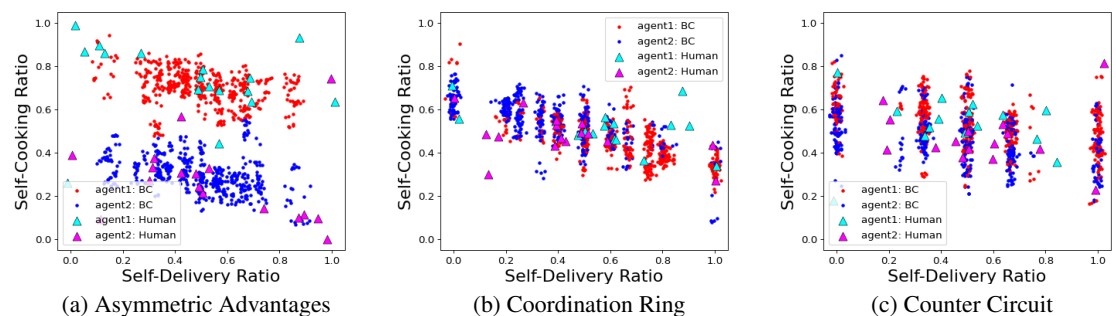

(a) Asymmetric Advantages      (b) Coordination Ring      (c) Counter Circuit

Figure 8: Trajectories induced by the learned human models and human players in Asymmetric Advantages, Coordination Ring and Counter Circuit. Each point or triangle denotes a trajectory with the X-axis coordinate being the self-cooking ratio, which is the ratio of onions the player places in the pot to the total amount of placements in the trajectory, and the Y-axis coordinate being the self-delivery ratio, which is the ratio of deliveries given by the player to the total number of deliveries in the trajectory. Triangles and points denote trajectories induced by human players and learned human models, respectively. Different colors stand for different player indices. "BC" represents the learned human models, and "Human" denotes human players. Clearly, trajectories induced by the learned human models can not fully cover those by human players.

trajectories by measuring *self-delivery ratio*, i.e., the ratio of deliveries by the specific player to the total delivery number in a trajectory, and *self-cooking ratio*, which is the ratio of onions that the player places in the pot to the total pot placement number in a trajectory. The distributions of these trajectories are demonstrated in Fig. 8. From the figure, we can observe that the learned human models can not fully cover human behaviors. This suggests that evaluation results with the learned human models can not provide a comprehensive comparison among different methods.

| Event | Value |
|---|---|
| Optimal placement | 3 |
| Picking up a dish from dish dispenser | 3 |
| Picking up a ready soup from the pot | 5 |

Table 16: Reward shaping for Asymmetric Advantages, Coordination Ring and Counter Circuit in the second stage.

| Event | Value |
|---|---|
| Picking up a dish from dish dispenser | 3 |
| Picking up a ready soup from the pot | 5 |

Table 17: Reward shaping for Distant Tomato and Many Orders in the first stage.

## F.2 ABLATION STUDIES

### F.2.1 POOL SIZE

Table 21 shows the average episode reward on 3 layouts with different sizes of the final policy pool for training the adaptive policy. Since increasing the pool size to 72 gives little improvement as suggested by the result, we use 36 in our experiments for computation efficiency.

### F.2.2 POLICY POOL CONSTRUCTION

HSP has two techniques for the policy pool, i.e., (1) policy filtering to remove duplicated biased policies and (2) the use of MEP policies under the game reward for half of the pool size. We measure the performance with human proxies by turning these options off. For "*HSP w.o. Filtering*", we keep all the policies by random search in the policy pool, which results in a larger pool size of 54 (18 MEP policies and a total of 36 random search ones). For "*HSP w.o. MEP*", we exclude MEP policies in the policy pool and keep all the biased policies without filtering, which leads to the same pool size of 36. The results are shown in Table. 22.

## F.3 COOPERATION WITH SCRIPTED POLICIES WITH STRONG BEHAVIOR PREFERENCES

Table 23 illustrates average episode reward and standard deviation (over 5 seeds) in all layouts with scripted policies. All values within a difference of 5 from the maximum value are marked in bold.

## F.4 HUMAN-AI EXPERIMENT

### F.4.1 EXPERIMENT SETTING

We recruited 60 volunteers by posting the experiment advertisement on a public platform. They are provided with a detailed introduction to the basic gameplay and the experiment process. The Overcooked game was deployed remotely on a server that the volunteers could access with their browsers. According to the feedback, over 90 percent of volunteers had no prior experience with Overcooked. We uniformly divided 60 volunteers into 5 groups assigned to each of the 5 layouts. We designed the experiment to last around 30 minutes for each volunteer to ensure the validity of the data. Due to availabilities of volunteers, experiments are conducted within two consecutive days. The experiment has two stages. In the first stage, which is called the *warm-up* stage, the participants are encouraged to explore the behaviors of 4 given AI agents without a time limit. After the first stage, they are required to comment on their game experience, e.g., whether the AI agents are cooperative and comfortable to play with, and rank the agents accordingly. In the second stage, each participant is instructed to achieve scores as high as they could in 24 games (4 AI agents $\times$ 2 player positions $\times$ 3 repeats).

We remark that, on the environment side, different from human-AI experiments performed by prior works (Zhao et al., 2021; Carroll et al., 2019) in Overcooked, we slow down the AI agents so that the AI agents have similar speed with human players. More specifically, 7 idle steps are inserted before each step of the AI agent. Such an operation is necessary since, in our prior user studies, we find that human players commonly feel uncomfortable if the AI agent is much faster and human players could contribute little to the score.

| Event | Value |
|-------|-------|
| Picking up a dish from the dish dispenser | 3 |
| Picking up a ready soup from the pot | 5 |

Table 18: Reward shaping for Many Orders in the second stage.

| Event | Value |
|-------|-------|
| Picking up a dish from the dish dispenser | 3 |
| Picking up a ready soup from the pot | 5 |
| Useful tomato pickup | 10 |
| Optimal tomato placement | 5 |
| Placing a tomato into an empty pot | -15 |

Table 19: Reward shaping for Distant Tomato in the second stage.

### F.4.2 HUMAN FEEDBACK

We collected and analyzed the feedback from the participants to see how they felt playing with AI agents. Here we summarize the typical reflections.

1. In *Coordination Ring*, the most annoying thing reported is players blocking each other during movement. To effectively maneuver in the ring-like layout, players must reach a temporary agreement on either going clockwise or counterclockwise. HSP is the only AI able to make way for the other player, while others can not recover by themselves once stuck. For example, both FCP and TrajDiv players tend to take a plate and wait next to the pot immediately after one pot is filled. But they can neither take a detour when blocked on their way to the dish dispenser nor yield their position to the human player trying to pass through. The video recorded in the human study can be found in Part 4.2 of `https://sites.google.com/view/hsp-iclr`.

2. In *Counter Circuit*, one efficient strategy is passing onion via the counter in the middle of the room: a player at the bottom fetches onions and places them on the counter, while another player at the top picks up the onions and puts them into pots. HSP is the only AI player capable of this strategy in both top and bottom places and performs the highest onion passing frequency cooperating with human players as shown in Figure. 6b.

3. In *Distant Tomato*, one critical thing is that mixed (onion-tomato) soups give no reward, which means two players need to agree on the soup to cook. All AI agents perform well when the other player focuses on onion soups. However, all AI agents except for HSP fail to deal with tomato-preferring partners as shown in Table. 3. FCP, MEP, or TrajDiv agents never actively choose to place tomatoes and keep placing onions even when a pot has tomatoes in it, resulting in invalid orders. On the contrary, HSP chooses to place tomatoes when there are tomatoes in the pot. Participants commonly agree that the HSP agent is the best partner to play with in this layout. The video recorded in the human study can be found in Part 4.2 of `https://sites.google.com/view/hsp-iclr`.

4. In *Many Orders*, most participants claim that HSP is able to pick up soups from all three pots, while other AI agents only concentrate on the pot in front of them and ignore the middle pot even if the human player attempts to use it. Table. 4 shows that HSP agent picks up most soups from the middle pot and meanwhile gets the highest average episode reward.

### F.4.3 HUMAN PREFERENCE ON DIFFERENT AI AGENTS

Figure. 9 illustrates human preference for different AI agents. In all layouts except a relatively restricted and simple layout, *Coordination Ring*, human players strongly prefer HSP over other AI agents. In *Coordination Ring*, though human players rank MEP above HSP, HSP is still significantly better than FCP and TrajDiv.

**Calculation Method**: Human preference for different methods is computed as follows. Assume we are comparing human preference between method A and method B. Let $N$ be the total number of human players attending the experiments in one layout, $N_A$ be the number of human players who

rank A over B, and $N_B$ be the number of those who rank B over A. "Human preference for method A over method B" is computed as $\frac{N_A}{N} - \frac{N_B}{N}$.

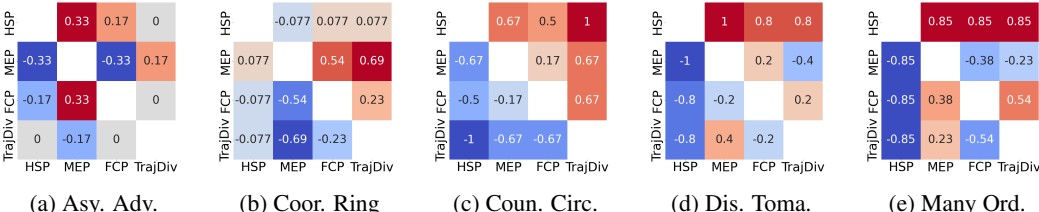

(a) Asy. Adv.     (b) Coor. Ring     (c) Coun. Circ.     (d) Dis. Toma.     (e) Many Ord.

Figure 9: Human preference for row partner over column partner in all layouts.

### F.4.4 SCORES IN THE SECOND STAGE

Table 24 shows average *reward per episode* during the second stage in all layouts. All methods have comparable episode rewards in Asymm. Adv and Coord. Ring. There is no room for improvement since all the methods have reached the highest possible rewards. In Counter Circ., the most complex layout in this category, HSP achieves a better performance than baselines: HSP has a 155+ reward while the most competitive baseline MEP has a reward of 134+. We remark that the reward difference between HSP and MEP is around 20, which is exactly the value of 1 onion soup delivery. This implies that the HSP agent can, on average, deliver one more soup than all the baselines per game episode with humans, which is a significant improvement.

|  | Pos. | Asy. Adv. | Coor. R. | Coun. Circ. |
|---|---|---|---|---|
| FCP | 1 | 282.8(9.4) | **161.3**(1.6) | 95.9(2.0) |
|  | 2 | 203.8(8.2) | **161.0**(2.7) | 92.7(1.3) |
| MEP | 1 | 291.7(4.6) | **161.8**(0.7) | **108.8**(4.2) |
|  | 2 | 203.4(2.0) | **164.2**(2.1) | **111.1**(0.7) |
| TrajDiv | 1 | 289.3(8.8) | 150.8(3.1) | 60.1(5.0) |
|  | 2 | 194.2(0.7) | 142.1(2.3) | 53.7(12.4) |
| HSP | 1 | **300.3**(2.2) | **160.0**(2.6) | **107.4**(3.5) |
|  | 2 | **217.1**(3.3) | **160.6**(3.3) | **106.6**(3.0) |

Table 20: Average episode reward and standard deviation (over 5 seeds) on 3 layouts for different methods played with human proxy policies. All values within 5 standard deviations of the maximum episode return are marked in bold. The Pos. column indicates the roles played by AI policies.

|  | Pos. | Asy. Adv. | Coor. R. | Coun. Circ. | Asy. Adv. | Coor. R. | Coun. Circ. |
|---|---|---|---|---|---|---|---|
|  |  | policy pool size = 36 | | | policy pool size = 72 | | |
| FCP | 1 | 282.8(9.4) | 161.3(1.6) | 95.9(2.0) | 278.3(16.0) | 158.9(0.6) | 91.9(7.5) |
|  | 2 | 203.8(8.2) | 161.0(2.7) | 92.7(1.3) | 200.9(13.2) | 156.9(4.7) | 90.7(4.8) |
| MEP | 1 | 291.7(4.6) | 161.8(0.7) | 108.8(4.2) | 298.2(5.4) | 157.3(2.7) | 104.6(5.0) |
|  | 2 | 203.4(2.0) | 164.2(2.1) | 111.1(0.7) | 207.8(7.3) | 158.9(3.0) | 105.0(2.2) |
| TrajDiv | 1 | 289.3(8.8) | 150.8(3.1) | 60.1(5.0) | 270.8(2.5) | 142.5(2.8) | 70.1(6.7) |
|  | 2 | 194.2(0.7) | 142.1(2.3) | 53.7(12.4) | 192.8(8.7) | 137.3(4.9) | 63.8(8.2) |

Table 21: Average episode reward and standard deviation (over 5 seeds) on 3 layouts for different methods played with human proxy policies. The Pos. column indicates the roles played by AI policies.

|  | Pos. | Asy. Adv. | Coor. R. | Cou. Circ. |
|---|---|---|---|---|
| HSP w.o. MEP (pool size = 36) | 1 | 308.5(4.4) | 157.5(3.0) | 94.0(2.7) |
|  | 2 | 219.6(15.9) | 157.7(2.5) | 100.4(1.1) |
| HSP w.o. Filtering (pool size = 54) | 1 | 311.3(8.1) | 139.2(5.6) | 80.1(4.6) |
|  | 2 | 209.3(4.0) | 138.5(3.1) | 88.7(0.9) |
| HSP (pool size = 36) | 1 | 300.3(2.2) | 160.0(2.6) | 107.4(3.5) |
|  | 2 | 217.1(3.3) | 160.6(3.3) | 106.6(3.0) |

Table 22: Average episode reward and standard deviation (over 5 seeds) on 3 layouts for different methods played with human proxy policies. The Pos. column indicates the roles played by AI policies.

| | Scripts | FCP | | MEP | | TrajDiv | | HSP | |
|---|---|---|---|---|---|---|---|---|---|
| | | 1 | 2 | 1 | 2 | 1 | 2 | 1 | 2 |
| Asy. Adv. | Oni. Pla. | 304.0(20.8) | 365.7(5.3) | 295.0(25.7) | 365.9(2.8) | 296.5(25.2) | 350.7(8.9) | **357.3**(2.4) | **396.3**(17.4) |
| | Oni. Pla.&Deli. | 364.9(2.9) | **230.5**(3.9) | **366.8**(1.5) | **230.3**(5.3) | 358.2(6.0) | 221.8(3.4) | **371.4**(3.3) | 229.9(4.9) |
| Coor. Ring | Oni. Every. | 106.0(8.7) | 112.1(7.2) | **124.6**(3.6) | **123.4**(3.2) | 115.7(11.1) | 118.1(6.7) | **120.8**(14.3) | **121.7**(10.9) |
| | Dish Every. | 91.9(4.1) | 97.0(3.4) | 101.4(3.1) | 99.0(7.3) | 107.5(5.2) | 107.1(5.3) | **114.9**(7.7) | **116.0**(7.2) |
| Coun. Circ. | Oni. Every. | 64.80(8.8) | 62.7(9.5) | 90.6(3.9) | 87.4(6.2) | 86.0(12.1) | 78.1(13.4) | **110.5**(3.7) | **104.4**(3.4) |
| | Dish Every. | 57.9(5.3) | 56.1(5.2) | 53.5(1.1) | 52.5(2.4) | 59.3(2.9) | 55.0(1.5) | **79.3**(3.5) | **77.8**(4.7) |
| | Oni.2Mid. Cou. | 73.0(11.5) | 72.4(12.6) | 112.1(6.5) | 110.7(5.8) | 109.5(5.6) | 107.9(5.5) | **149.3**(3.1) | **147.1**(3.2) |
| Dis. Toma. | Toma. Pla. | 13.5(2.6) | 17.8(7.6) | 25.9(17.6) | 14.3(3.6) | 29.6(14.1) | 17.2(4.8) | **277.7**(9.7) | **278.9**(18.8) |
| | Toma. Pla.&Deli. | 197.5(4.9) | 158.3(7.3) | 185.9(9.9) | 174.8(7.4) | 175.3(21.6) | 154.3(17.6) | **237.2**(8.2) | **231.9**(22.0) |
| Many Ord. | Toma. Pla. | 285.4(19.5) | 279.8(12.8) | 224.1(61.2) | 227.6(60.5) | 263.8(10.1) | 254.6(5.7) | **315.7**(12.1) | **319.9**(6.5) |
| | Toma. Pla.&Deli. | **334.1**(5.6) | **324.2**(4.9) | **329.1**(12.7) | **327.2**(12.4) | 298.8(3.2) | 292.6(1.5) | 326.7(4.0) | 322.3(3.9) |

Table 23: Average episode reward and standard deviation (over 5 seeds) in all layouts with scripted policies. All values within Within a difference of 5 from maximum reward are marked in bold. The "1" and "2" indicates the roles played by AI policies.

|  | Pos. | Asy. Adv. | Coor. Ring | Cou. Circ. | Dis. Toma. | Many Ord. |
|---|---|---|---|---|---|---|
| FCP | 1 | **339.3**(38.17) | 185.0(19.73) | 127.7(28.14) | **351.3**(82.25) | 312.4(58.73) |
|  | 2 | 321.3(34.80) | 180.7(22.98) | 118.3(29.20) | **320.5**(66.49) | 321.3(61.12) |
| MEP | 1 | 329.7(45.97) | **193.3**(22.11) | 136.9(27.00) | 341.9 (65.07) | 322.0(50.53) |
|  | 2 | **324.2**(39.93) | 183.6(26.75) | 134.5(28.63) | 313.9(78.29) | 319.2(52.98) |
| TrajDiv | 1 | 329.0(43.18) | 184.7(28.60) | 112.6(25.78) | 327.1(71.04) | 312.9(62.82) |
|  | 2 | 318.8(48.97) | 176.8(31.33) | 105.0(31.05) | 316.0(77.65) | 334.2(57.99) |
| HSP | 1 | 336.0(35.55) | 185.5(38.92) | **158.0**(28.56) | 331.6(61.33) | **384.3**(47.50) |
|  | 2 | 318.8(48.97) | **188.9**(22.00) | **155.2**(23.43) | 305.9(58.61) | **380.7**(62.27) |

Table 24: Average reward per episode in all layouts with human players in the second stage.

