# OpenReview forum: "Learning Zero-Shot Cooperation with Humans, Assuming Humans Are Biased"
_ICLR.cc/2023/Conference — ICLR 2023 poster_

### Official Review · Reviewer_gDCQ · 2022-10-21

**Confidence:** 4
**Correctness:** 4
**Technical Novelty And Significance:** 3
**Empirical Novelty And Significance:** 3
**Recommendation:** 6

**Clarity, Quality, Novelty And Reproducibility:**

The clarity of this paper is good and every claim is well justified.

The novelty of this paper is good, mainly depending on the novel practical research problem I have never seen before. Nevertheless, I have to say that adaption to the unknown agent has a long history of research in the multi-agent learning. The research problem discussed in this paper can be seen as a special case, where an agent is replaced by human being.

The originality of this paper is good, because of the novel research problem and the experimental design.

The reproducibility of this paper is generally good, since it provides the details of experiments. However, I am concerned about the experiment done with human beings, which could be difficult to be reproduced.

Overall, the quality of this paper is high.

**Strength And Weaknesses:**

## Strength
1. This paper is well written and easy to follow. The related works are sufficiently discussed.
2. The experimental settings and the baseline algorithms are described in details.
3. Although this paper is an application work, it still includes some theoretical analysis as the base to support the proposed algorithm. The theoretical analysis is neat and sufficient to support the idea.
4. No obvious flaws in my view.


## Weaknesses
1. The construction of the hidden reward function is implemented as the counting-based method which is too simple. Since the hidden reward function is belonging to a type of intrinsic reward, there are plentiful literatures that study on the construction or learning this reward. I suggest the authors can have a review on these and improve the work.
2. In the procedure of policy filtering, the greedy policy selection suffers computational burdern as the authors illustrate. I wish the authors give a discussion on the possible way to improve it in the future work, since I don't wish the performance improvement is just based on more computation which is not good for the following works.

**Summary Of The Paper:**

This paper proposes a novel method called Hidden-Utility Self-Play (HSP) to solve a realistic and challenging problem that there exist human biases in human-AI interaction. The main contribution of this paper is proposing a hidden reward function to model human biases. The experiments are solid and sufficient to verify the performance improvement of HSP.

**Summary Of The Review:**

This paper proposes a novel practical research problem and a novel algorithm to solve it. Besides, it has done solid and sufficient experiments to verify the performance of the proposed algorithm.

---

> ### Author Response · Authors · 2022-11-13
> **Thanks for the review. We have updated the future work according to the feedback.**
>
> Thanks for the constructive review. We have revised the paper to have a more detailed discussion on future work. Here are our responses:
>
> ### Q1: The construction of the hidden reward function is implemented as the counting-based method which is too simple. Since the hidden reward function belongs to a type of intrinsic reward, there is plentiful literature that is studied on the construction or learning this reward. I suggest the authors can have a review on these and improve the work.
>
> **A1**:  We thank the reviewer for the valuable review. We remark that we discuss how to construct the reward function in the last paragraph of the related work, mainly focusing on the event-based intrinsic reward design in the past literature. There are also works on the more general paradigm of learning intrinsic rewards rather than assuming a suitable set of events [1, 2]. We have updated the future work to have a discussion on these works.
>
> ### Q2: In the procedure of policy filtering, greedy policy selection suffers computational burden as the authors illustrate. I wish the authors give a discussion on the possible way to improve it in future work, since I don't wish the performance improvement is just based on more computation which is not good for the following works.
>
> **A2**:  Thanks for the feedback. We acknowledge that the computation expense paid to obtain a diverse pool of policies is one major limitation of HSP. Techniques like bootstrapping new policies from old ones could help reduce the training cost. For example, we can learn a new policy based on Successor Feature (SF)[3], which is a value function representation that decouples the dynamics of the environment from the rewards. We can first train several basic policies and then use SF to construct a diverse policy pool based on these basic policies. This solution can significantly reduce the computation cost, but it also introduces new problems, such as how to choose a sufficiently diverse set of basic policies. We leave this as our future work.
>
> **References:**
>
> [1] Zihan Zhou, Wei Fu, Bingliang Zhang and Yi Wu, "Continuously Discovering Novel Strategies via Reward-Switching Policy Optimization", International Conference on Learning Representations, 2022.
>
> [2] Shah, Rohin, Noah Gundotra, Pieter Abbeel, and Anca Dragan. "On the feasibility of learning, rather than assuming, human biases for reward inference." In International Conference on Machine Learning, pp. 5670-5679. PMLR, 2019.
>
> [3] Barreto, A., Dabney, W., Munos, R., Hunt, J.J., Schaul, T., van Hasselt, H.P. and Silver, D., 2017. Successor features for transfer in reinforcement learning. Advances in neural information processing systems, 30.

---

> > ### Comment · Reviewer_gDCQ · 2022-11-18
> > **Thank you for your response**
> >
> > Dear Authors,
> >
> > Thank you for your response. It has addressed my concerns.

---

### Official Review · Reviewer_ufwf · 2022-10-25

**Confidence:** 4
**Correctness:** 4
**Technical Novelty And Significance:** 3
**Empirical Novelty And Significance:** 3
**Recommendation:** 6

**Clarity, Quality, Novelty And Reproducibility:**

In terms of clarity/quality, it would be helpful to clarify the following points:
- It would be beneficial to more clearly point the reader to where "Asylum. Adv.", "Coord. Ring", and "Counter Circ." are specified in the appendix so that they may better follow the results and take aways. I missed the pointer to the appendix and it made following the results and take aways more challenging.
- It was unclear to me exactly how the target policy's adaptation occurs. Is there any adaptation happening online while the policy is interacting with one of the humans? Or is the policy "adaptive" because it has experienced co-play with agents trained according to a variety of different reward functions? Is the contribution only to train the policy pool on random reward functions?
- It would be nice have it made clear what the criteria for a "fair comparison" (6. Experiments | Baselines), which is used to motivate the pool size.
- Why was 5 standard deviations used instead of 1 when assessing performance? (6. Experiments | Baselines)

In terms of novelty:
- The paper builds upon existing methods, but modifies how the pool of policies is trained. Based on the given literature review, the introduction of random rewards for policy pool training appears novel. However, I am not an expert in self-play and fictitious co-play.

In terms of reproducibility:
- The authors have not made code available and it would be helpful for them to release their code.
- The appendix has a lot of implementation details and looks sufficient for reproducibility. However I did not attempt to reproduce and have not previously implemented the baselines, so cannot definitively say if sufficient information is given.


**Strength And Weaknesses:**

Strengths:
- Overall a well written paper.
- The method is well motivated for why it is needed and is well grounded in the existing literature.
- The experiments appear to be solid and compare against important baselines to evaluate different aspects of the proposed contributions. The authors made sure to explicitly evaluate how well the approach works when the target policy is deployed to collaborative with agents that have reward functions not included in the policy training pool. Additionally, the authors evaluate against a variety of "humans", including real humans. Assessing the method across different "humans" provides a good idea of how the method extends to the real world.
- The use of random reward search to create a diverse pool of policies seem simple, but also effective.
- The authors discuss limitations of the current approach and propose future work to address the limitations.

Weaknesses:
- It would be nice to see a discussion about what the ablation study results tell us about the proposed method. What does it mean for the MEP policies to be helpful? What does it mean about the method for a larger batch size to be important? How does the need for a large batch size align with what the baseline methods needs?
- It is unclear to me how relevant the paper is to the topic areas ICLR focuses on. There is little to no discussion of representations nor representation learning methods. The paper seems like it is a better fit for CoRL, AAAI, AAMAS, and ICML.

**Summary Of The Paper:**

The paper presents a method for learning a collaborative agent that is able to handle arbitrary and/or suboptimal human reward functions. The method leverages self-play, specifically fictitious co-play, where the policies the target policy plays alongside are learned according to different reward functions. A variety of reward functions are used to train the pool of policy in order to train the target policy alongside a variety of different preference types and to teach the target policy to be robust to different, potentially previously unseen, preferences. The proposed method is evaluated on the Overcooked task, a standard benchmark for human-agent collaboration, alongside learned human models, scripted policies, and real humans. Across the different "humans", the proposed method outperforms the baselines. The experiments ablate several aspects of the approach: how the policies in the pools are trained and the batch size used to train the target policy. The results suggest that both filtering and MEP policies (trained with a single reward function) trained policies are important and large batch sizes are needed to stable target policy performance.

**Summary Of The Review:**

Overall the paper is strong with some areas for improvement in terms of clarity. The idea is relatively simple and based on the results appears to improve policy performance, especially in experimental conditions including real humans. My only concern about acceptance to ICLR is its fit. The paper seems like a better fit to CoRL, AAAI, AAMAS, or ICML as it does not have a representation nor representation learning focus.

---

> ### Author Response · Authors · 2022-11-13
> **Part 2: Thanks for the review. We have revised the paper to address the reviewer's concern.**
>
> **References:**
>
> [1] Shih, Andy, Arjun Sawhney, Jovana Kondic, Stefano Ermon, and Dorsa Sadigh. "On the critical role of conventions in adaptive human-AI collaboration." ICLR, 2021.
>
> [2] Puig, Xavier, Tianmin Shu, Shuang Li, Zilin Wang, Yuan-Hong Liao, Joshua B. Tenenbaum, Sanja Fidler, and Antonio Torralba. "Watch-and-help: A challenge for social perception and human-ai collaboration." ICLR spotlight, 2021.
>
> [3] Li, Quanyi, Zhenghao Peng, and Bolei Zhou. "Efficient Learning of Safe Driving Policy via Human-AI Copilot Optimization." ICLR, 2022.

---

> ### Author Response · Authors · 2022-11-13
> **Part 1: Thanks for the review. We have revised the paper to address the reviewer's concern.**
>
>
>
>
>
> We thank the reviewer for the detailed review. We have revised our paper to address the concern about the ablation study and clarify the pointer&criteria in the main paper.
>
> ### Q1: It would be nice to see a discussion about what the ablation study results tell us about the proposed method. What does it mean for the MEP policies to be helpful? What does it mean about the method for a larger batch size to be important? How does the need for a large batch size align with what the baseline methods need?
>
> **A1**: Thanks for the review. We have revised the paper to have a discussion about the ablation study. The MEP policies are trained with a cross-entropy bonus to improve the diversity of the pool. We find that including some diversity-enhanced policies in the pool can be effective in practice, and thus include MEP policies in the policy pool of HSP for the best empirical performance. We also emphasize that only using diversity-enhanced policies to build up the pool is not sufficient, as shown in the experiment results.  Regarding the question about the batch size, we point out that most existing baselines use a fairly small batch size. However, we notice that when using a larger batch size, even the most basic baseline can be very competitive. Therefore, we use a sufficiently large batch size for a fair comparison. We remark that we re-implement all baselines and the performances of the baselines are better than the original papers. More details about the baselines can be found in Appendix D.2.
> ### Q2: It is unclear to me how relevant the paper is to the topic areas ICLR focuses on. There is little to no discussion of representations nor representation learning methods. The paper seems like it is a better fit for CoRL, AAAI, AAMAS, and ICML.
>
> **A2**: Human-AI Coordination is a fundamental problem for the multi-agent system, and also a sub-topic of Multi-Agent Reinforcement Learning (MARL) and General Artificial Intelligence (GAI). Our purpose is to build an adaptive agent for human-compatible AI, which, according to the call-for-paper page, belongs to the critical subject area of reinforcement learning covered by ICLR. There are also several papers focused on human-AI Coordination that have been published in ICLR [1,2,3], therefore we believe this work is relevant to ICLR. Besides, we would like to clarify that although the conference name is ICLR (International Conference on Learning Representations), in recent years it has been receiving papers on topics of general deep learning with a broad range of subject areas rather than a narrow subject area of representation learning.
>
> ### Q3: It was unclear to me exactly how the target policy's adaptation occurs. Is there any adaptation happening online while the policy is interacting with one of the humans? Or is the policy "adaptive" because it has experienced co-play with agents trained according to a variety of different reward functions? Is the contribution only to train the policy pool on random reward functions?
>
> **A3**:  A policy produced by HSP is expected to be adaptive because:
> - The target policy is trained with a diverse policy pool, which covers as many policies as possible that are similar to real human behavior.
> - We use an RNN in the network architecture so that the agent implicitly performs online adaptation at test time. To be more explicit, the target policy does not know the teammate's or opponent's strategy in each episode, so it is forced to adapt, and we use an RNN-based neural network policy to generalize to unseen partners.
>
> We remark that both techniques are important.
>
> Regarding the contribution, we summarize as follows:
> - We demonstrate the importance of human preference in human-AI coordination and empirically verified the advantages of our proposed method.
> - We conducted thorough experiments and analysis which provided more insights into the literature, while most prior works are limited to using game scores as the only evaluation metric.
>
> ### Q4: Why was 5 standard deviations used instead of 1 when assessing performance?
>
> **A4**: We first apologize for the confusion. Maximum returns or comparable returns within a threshold of 5 are marked in bold. We chose 5 because the final task score is calculated as the number of deliveries times 20 points, where 1 point is too small to reflect the difference between methods.
>
> ### Q5: The authors have not made code available and it would be helpful for them to release their code.
> **A5**: We have submitted our code in the supplementary material (fold named code) and also on our website: https://sites.google.com/view/hsp-iclr (button named Code). Please let us know if there is anything wrong.

---

> > ### Comment · Reviewer_ufwf · 2022-11-18
> > **Clarification Question on A3**
> >
> > To make sure I am understanding correctly, the approach is adaptive in terms of the strategy the policy is able to deploy and once the policy is learned, its parameters are fixed?
> >
> > Thank you!

---

> > > ### Author Response · Authors · 2022-11-18
> > > **Thanks for the feedback. Here is our response:**
> > >
> > > Thanks for the feedback. The parameters of the policy are fixed once the HSP policy is trained and ready to deploy. As we clarified previously, our approach is adaptive by using an RNN in the network architecture. Although the parameters of the policy are fixed during interactions with other agents or humans, the hidden state of the policy changes according to the actions of the co-partner.

---

### Official Review · Reviewer_hpUU · 2022-10-25

**Confidence:** 4
**Correctness:** 3
**Technical Novelty And Significance:** 3
**Empirical Novelty And Significance:** 3
**Recommendation:** 6

**Clarity, Quality, Novelty And Reproducibility:**

This paper shows the effectiveness of a variation on a popular technique in which there is a hidden reward function to augment training, expands the Overcooked domain with two new layouts, and shows the effectiveness in human studies; these are all reasonably novel contributions. There are adequate details for reproducibility (although, as mentioned above, more details on the human study should be included, possibly in an expanded Appendix with some detail in the main paper).

**Strength And Weaknesses:**

Strengths
- This paper focuses on the benefits of incorporating human biases in cooperating with them, an important area.
- The paper is written clearly, and the appendix incorporates many details of the experiments that aid reproducibility.
- The ablation studies and other experimental details are presented in a thorough way.

Weaknesses
- It is not clear how well the approach presented here would generalize and requires domain knowledge to be effective.
- The human study, which shows the most compelling evidence for the effectiveness of this framework, needs more details. How were the volunteers recruited, how did they participate in the experiment, how long did the experiments take, what was their experience with Overcooked, etc.? The volunteer group is also younger and heavily male. How may this impact the results?
- The "Distant Tomato" regime has a clear optimal policy - would just telling the human participants what that is have substantially impacted results? Specifically, is this a question of a "hidden reward function" (as the framework presents) or inadequate skill or knowledge of the domain by the human participants?

**Summary Of The Paper:**

This paper proposes the Hidden-Utility Self-Play algorithm to explicitly model human bias via a modification to the reward function used during self-play training. This is used in the domain of the cooperative game Overcooked, where a particular reward modification is shown to be effective both in experiments where it is paired with learned human models and in experiments with actual humans. The framework is demonstrated to be especially effective with real human partners.

**Summary Of The Review:**

This is an interesting paper that shows the effectiveness of incorporating models of human biases in a particular domain. The main weaknesses are the question of how domain-specific this reward is (and how much domain knowledge it requires) and inadequate details about the human study. For example, in Section F.4.1, slowing down the speed of the simulation is noted for making the users more comfortable in contributing; these are the kinds of factors that could substantially impact the effectiveness of a solution, in addition to algorithmic contributions. The qualitative details in F.4.2 could also be augmented with quantitative metrics of the feedback, if available.

---

> ### Author Response · Authors · 2022-11-13
> **Thanks for the review. We have revised the paper&appendix to address the reviewer's concern.**
>
> We sincerely thank the reviewer for the valuable review and feedback. We have added more details about human studies in the main paper and appendix. Our detailed responses are listed below.
>
> ### Q1: It is not clear how well the approach presented here would generalize and require domain knowledge to be effective.
>
> **A1**: Thanks for the feedback. We argue that our key insight is that in human-AI cooperation, modeling human biases could be very critical to building an assistive AI. Although we would need domain knowledge to design a new set of events or features when adapting to a new task, the algorithm is general to the domain. Also, we want to point out that there exist works that can learn biased behaviors without selecting a suitable set of events [1, 2], which we will leave as future work.
>
> ### Q2: The human study, which shows the most compelling evidence for the effectiveness of this framework, needs more details. How were the volunteers recruited, how did they participate in the experiment, how long did the experiments take, what was their experience with Overcooked, etc.? The volunteer group is also younger and heavily male. How may this impact the results?
>
> **A2**: Thanks for the constructive review. We have added more details of the human study both in the main paper and appendix. Please let us know if there is still ambiguity, and we will continue to improve the manuscript. Here we give a brief description.  We recruited volunteers by posting the human-AI experiment advertisements on the public platform. The game was deployed on a website and volunteers were able to finish the experiment on their own computers. Detailed game instructions were given to each volunteer to ensure they fully understood the rules of the game. Due to availabilities of volunteers, experiments are conducted within two consecutive days. We designed the experiment to last around 30 minutes for each volunteer to ensure the validity of the data. We also counted the volunteers' experience of the overcooked game, and we found that over 90% had no experience of playing the Overcooked game. We have made our best efforts to recruit volunteers. Unfortunately, they were mainly from the engineering department, leading to younger and more males. The experiment results still can reflect the differences between methods, as the score is more relevant to the volunteers' game experience. Therefore, we designed a two-stage experiment scheme, including a warm-up stage to become familiar with the game and an exploration stage to obtain scores as high as possible.
>
> ### Q3: The "Distant Tomato" regime has a clear optimal policy - would just tell the human participants what that is have substantially impacted results? Specifically, is this a question of a "hidden reward function" (as the framework presents) or inadequate skill or knowledge of the domain by the human participants?
>
> **A3**: Thanks for the feedback. If we directly show the optimal policy to human participants, we are actually forcing or guiding the participants to follow the optimal policy, so their true preferences will be influenced, which is opposite to our goal. In this work, our focus is to build an AI to assist humans rather than forcing humans to follow the convention of the AI agent, i.e. the uni-modal optimal behavior. That's exactly the reason why we set up the experiments in two phases: a warm-up stage and an exploitation stage. In the warm-up stage, humans are expected to freely explore their preferences and test whether the AI can still be assistive. In the exploitation stage, we follow the traditional experiment setting where the goal of humans and the agent is to maximize the task reward. Interestingly, we find that even when the goal of maximizing the task reward is clearly given, human behaviors still show some diversity (Fig.6, Tab.3 and Tab.4 in Sec.6.4.2), which, we believe, can be served as a better evaluation protocol.
>
> **References:**
>
> [1] Zihan Zhou, Wei Fu, Bingliang Zhang and Yi Wu, "Continuously Discovering Novel Strategies via Reward-Switching Policy Optimization", International Conference on Learning Representations, 2022.
>
> [2] Shah, Rohin, Noah Gundotra, Pieter Abbeel, and Anca Dragan. "On the feasibility of learning, rather than assuming, human biases for reward inference." In International Conference on Machine Learning, pp. 5670-5679. PMLR, 2019.

---

### Official Review · Reviewer_SZ9c · 2022-10-28

**Confidence:** 4
**Correctness:** 4
**Technical Novelty And Significance:** 2
**Empirical Novelty And Significance:** 4
**Recommendation:** 6

**Clarity, Quality, Novelty And Reproducibility:**

The paper is clearly written with significant details available in the appendix. The work is original. The units in Figure 5 should be explained in the text


**Strength And Weaknesses:**

The empirical results are excellent. In particular, I think the combination of ablations, human experiments, scripted bots, and imitation-trained policies go beyond most any other works and clarify key issues that were not carefully analyzed in previous works. For instance, the authors clearly show that the techniques used in most evaluations (imitation-learned policies and human interaction) are highly confounded since people adapt, and the imitation-learned policies don’t show much diversity. Their method only weakly improves over baselines in these tasks. In contrast, the use of specific scripted probes and a more qualitative evaluation revealed large discrepancies between actual coordination performance in the most important edge cases.

The algorithmic contribution is a weakness as it depends on significant hand-tuning of custom features specific for these specific Overcooked environments. I do not see how this approach could be easily adapted to a new tasks (or even an Overcooked level with different dynamics). On its own, I do not think this algorithm is a sufficient contribution to literature. I would have also liked to see comparisons or thoughts on more model-based towards generating diversity in Overcooked for example: Wu, Sarah A., et al. "Too Many Cooks: Bayesian Inference for Coordinating Multi‐Agent Collaboration." Topics in Cognitive Science 13.2 (2021): 414-432.



**Summary Of The Paper:**

The authors train agents with self play that have diverse preferences that differ from the original task and show that having a richer pool of agents generated with this technique yields policies that are better adapted to human coordination.


**Summary Of The Review:**

Based on the new method alone I would not accept this paper. However, the thoroughness of evaluation sets a new standard and I feel that I learned something new and important from these empirical analyses. I would like to cite this paper in the future and that should be sufficient for acceptance. I would raise my score further if the authors can more greatly emphasize these contributions in their work

---

> ### Author Response · Authors · 2022-11-13
> **Thanks for the review. We have revised the paper to address the reviewer's concern.**
>
> We thank the reviewer for the constructive review and feedback. We have revised the paper to clarify the issues in Fig.5 and contributions. Our detailed responses are listed below.
>
> ### Q1: I do not see how this approach could be easily adapted to a new task (or even an Overcooked level with different dynamics). On its own, I do not think this algorithm is a sufficient contribution to literature.
>
> **A1**: Thanks for the feedback. To adapt HSP to a new task or a new domain, we should select a new set of features or events, which is very similar to conventional feature engineering and reward shaping, while the algorithm framework remains unchanged. Notably, designing a set of events is much easier than handcrafting a set of policies, which are much higher dimensional. We also remark that the contribution of our work is to argue that modeling human biases can be critical to building assistive AI in human-AI cooperation with deep reinforcement learning, which has not been much explored in MARL literature.
>
> ### Q2: I would have also liked to see comparisons or thoughts on more model-based towards generating diversity in Overcooked for example: Wu, Sarah A., et al. "Too Many Cooks: Bayesian Inference for Coordinating Multi‐Agent Collaboration." Topics in Cognitive Science 13.2 (2021): 414-432.
>
> **A2**: [1] utilizes Bayesian Delegation, a model-based method, to enable agents to rapidly infer the hidden intentions of others by inverse planning, which is also heavily based on domain knowledge. Our approach HSP first constructs a diverse policy pool to cover as many behavior modalities as possible and then trains an adaptive policy against the diverse policy pool. Both methods focus on the task of zero-shot coordination without human data. While [1] requires constructing sub-tasks, which may not cover a sufficient variety of behavioral modalities, HSP does not require constructing sub-tasks and only needs random sampling to build a diverse policy pool. The focus of [1] and HSP are orthogonal since [1] focuses more on the adaptive stage while HSP aims to find more diverse strategies. Besides, [1] requires explicit bayesian inference while HSP trains an adaptive agent in an end-to-end fashion, which is more general.
>
> **References:**
>
> [1] Wu, Sarah A., et al. "Too Many Cooks: Bayesian Inference for Coordinating Multi‐Agent Collaboration." Topics in Cognitive Science 13.2 (2021): 414-432.

---

> > ### Comment · Reviewer_SZ9c · 2022-12-03
> > **Thank you**
> >
> > Thank you for the response. Please include the discussion of model-based methods in the final version of the paper.

---

> > > ### Author Response · Authors · 2022-12-03
> > > **We have updated the paper to have a discussion in the related work.**
> > >
> > > We thank the reviewer for the feedback, we have updated the paper to have a discussion on model-based methods in the related work. Here is the updated paper [link](https://drive.google.com/drive/folders/1eqlDHWlDF5_YwJ_Kad4tMGudZgCoGS33) with changes in red.

---

### Author Response · Authors · 2022-11-13
**We have updated the paper&appendix to address reviewers' concerns.**

We sincerely thank the valuable reviews and feedback from all reviewers. We have revised the **main paper and appendix** with updates marked in red. The main changes include:
- Contributions to the thoroughness of the evaluation
- More details on human study
- Issues in Fig.5 and the pointer
- Summary of the ablation study
- More discussion on future work

---

### Decision · Program_Chairs · 2023-01-20

**Decision:**

Accept: poster

**Justification For Why Not Higher Score:**

Approach is motivated by not needing data from human policies, but does assume some domain specific knowledge of likely human motivations.

Core problem likely of interest to only a sub-community within ICLR, arguably would reach a larger audience at other conferences.

**Justification For Why Not Lower Score:**

Unanimous support for the paper across all reviewers with a variety of contributions identified by all.

**Metareview: Summary, Strengths And Weaknesses:**

This paper explores how to accommodate for human biases when training agents for human-AI collaboration without data from human policies. The proposed method is demonstrated in Overcooked (a popular choice recently in studies of human-AI collaboration) but is in principle more general then this game alone provided domain-specific knowledge of events likely to motivate human behavior. This requirement of domain-specific knowledge weakens the original motivation and benefit of not needing data from human policies. All reviewers saw positive contributions from the paper suggesting the paper is of interest to ICLR attendees, but the authors may reach a larger audience by also promoting the work in more core AI venues.

**Note From Pc:**

if the above contains the word "oral" or "spotlight" please see: "oral" presentation means -> notable-top-5% and "spotlight" means -> notable-top-25%. As stated in our emails, we are disassociating presentation type from AC recommendations